# Preparation of Silicon Hydroxyapatite Nanopowders under Microwave-Assisted Hydrothermal Method

**DOI:** 10.3390/nano11061548

**Published:** 2021-06-11

**Authors:** Zully Matamoros-Veloza, Juan Carlos Rendon-Angeles, Kazumichi Yanagisawa, Tadaharu Ueda, Kongjun Zhu, Benjamin Moreno-Perez

**Affiliations:** 1Graduate Division, Technological Institute of Saltillo, Tecnológico Nacional de México/(I.T. Saltillo), Saltillo 25280, Mexico; bnmoreno24@gmail.com; 2Centre for Research and Advanced Studies of the NPI, Cinvestav-Campus Saltillo, Saltillo 25900, Mexico; jcarlos.rendon@cinvestav.edu.mx; 3Research Laboratory of Hydrothermal Chemistry, Faculty of Science, Kochi University, Kochi 780-8520, Japan; yanagi@kochi-u.ac.jp; 4Department of Marine Resources Science, Faculty of Agricultural and Marine Science, Kochi University, Nankoku 783-8502, Japan; chuji@kochi-u.ac.jp; 5Center for Advanced Marine Core Research, Kochi-University, Nankoku 783-8502, Japan; 6State Key Laboratory of Mechanics and Control Mmechanics Structures, Nanjing University of Aeronautics and Astronautics, Nanjing 210016, China; kjzhu@nuaa.edu.cn

**Keywords:** hydrothermal microwave assisted synthesis, silicon-hydroxyapatite, nano powders

## Abstract

The synthesis of partially substituted silicon hydroxyapatite (Si-HAp) nanopowders was systematically investigated via the microwave-assisted hydrothermal process. The experiments were conducted at 150 °C for 1 h using TMAS (C_4_H_13_NO_5_Si_2_) as a Si^4+^ precursor. To improve the Si^4+^ uptake in the hexagonal structure, the Si precursor was supplied above the stoichiometric molar ratio (0.2 M). The concentration of the TMAS aqueous solutions used varied between 0.3 and 1.8 M, corresponding to saturation levels of 1.5–9.0-fold. Rietveld refinement analyses indicated that Si incorporation occurred in the HAp lattice by replacing phosphate groups (PO_4_^3−^) with the silicate (SiO_4_^−^) group. FT-IR and XPS analyses also confirmed the gradual uptake of SiO_4_^−^ units in the HAp, as the saturation of Si^4+^ reached 1.8 M. TEM observations confirmed that Si-HAp agglomerates had a high crystallinity and are constituted by tiny rod-shaped particles with single-crystal habit. Furthermore, a reduction in the particle growth process took place by increasing the Si^4+^ excess content up to 1.8 M, and the excess of Si^4+^ triggered the fine rod-shaped particles self-assembly to form agglomerates. The agglomerate size that occurred with intermediate (0.99 mol%) and large (12.16 mol%) Si contents varied between 233.1 and 315.1 nm, respectively. The excess of Si in the hydrothermal medium might trigger the formation of the Si-HAp agglomerates prepared under fast kinetic reaction conditions assisted by the microwave heating. Consequently, the use of microwave heating-assisted hydrothermal conditions has delivered high processing efficiency to crystallize Si-HAp with a broad content of Si^4+^.

## 1. Introduction

The preparation of biomaterials with similar chemical and physical properties to biological hydroxyapatite (HAp), in terms of their chemical and physical properties, involves the uptake of cations and anions in the hexagonal HAp structure. The incorporation of Si^4+^ ions into the PO_4_^3−^ unit network of the HAp stimulates both bone formation and resorption processes, which are relevant to both tissue restauration and bone growth [1]. Furthermore, the incorporation of Si^4+^ has been incorporated simultaneously with calcium during the early stage of calcination [2]. Additionally, silicon is also essential for other biological soft tissue functionality, such as cartilage growth, and synthetic calcium-phosphate bioceramics containing low Si^4+^ contents in their structures, which had marked biological properties for bone restauration [3,4]. Hitherto, Si-HAp bioceramics were prepared with a conventional solid-state reaction at 1000 °C for 6 h, employing β-tricalcium phosphate (β-Ca_3_(PO_4_)_2_), silicon dioxide (SiO_2_), and calcium carbonate (CaCO_3_). These partially silicon-substituted powders exhibit a good biological dissolution capability in comparison with pure HAp [5].

Moreover, chemical solution methods, such as coprecipitation, neutralization, and sol-gel, are alternative synthetic procedures for producing both HAp and Si-HAp nanoparticles. The increase of silicon incorporation in the HAp structure provokes a marked decrease in the crystallite size [6,7,8,9,10,11]. Likewise, silicon affects the particle morphology during the embryo precipitation and particle growth processes. Recently, extensive attention has been paid to the appropriated process with a Si^4+^ precursor reagent to overcome the difficulties in handling associated with its reactivity under wet chemical processing [4,5,6,7,8,9,10,11,12,13,14,15].

Hitherto, tetraethyl orthosilicate (Si (OCH_2_CH_3_)_4_, TEOS) in polyethylene glycol/water and silicon tetra-acetate in water (Si (CH_3_CO_2_)_4_) have been better reagents for incorporating Si^4+^ in the hexagonal structure [13]. The maximum efficiency of the Si^4+^ incorporation in the apatite structure was 90% according to the nominal stoichiometric content of 8.0 mol%. Wet chemical quantitative analyses revealed the presence of silicon ions hydrolyzed in the remaining mother liquor after precipitation of SiHAp. In contrast, the challenge of producing synthetic Si-HAp has been carried out by various methods, including soft chemistry processes [5,6,7,8,9,10,11,12,13].

The slow reaction kinetic of the ions species to produce pure HAP and other solid solutions required prolonged processing time in specific systems [14,15,16,17,18].

The hydrothermal process has brought further advantages in terms of chemical reactivity: Higher yield for crystalline products with nanometric size and reaction kinetics enhancement even at relatively low temperatures (100–250 °C) [18,19].

Likewise, a few pioneering research works have reported the synthesis of partially substituted Si-HAp under hydrothermal conditions [12,14]. The synthesis was conducted by two pathways. The first experiments were conducted using the chemical reagents (NH_4_)_3_PO_4_ and TEOS as precursor of PO_4_^3−^ and SiO_4_^4−^ ions. However, the synthesis conducted at 200 °C for 8 h limited the incorporation to only 8.0 mol% Si^4+^, regardless of the nominal stochiometric amount intended (9.0 mol%) [13,14]. In other experiments, the uptake of Si^4+^ was further limited by using (NH_4_)_2_HPO_4_ to 7.65 mol%. The partially substituted Si-HAp particles also incorporate CO_3_^2−^ ions, and the presence of these ions is reported to hinder the incorporation of SiO_4_^4−^ during the crystallization and particle coarsening steps [13,20,21].

Similar experiments were recently conducted to attempted the synthesis of Si-HAp under hydrothermal conditions at 150 °C for 10 h by employing tetramethyl ammonium silicate ((C_4_H_13_NO_5_Si_2_), TMAS) [1,2,3,13,14,15,16,18,19,20]. The low silicon reactivity in the hydrothermal alkaline medium at a pH of 10 caused a limited Si^4+^ content in the HAp structure of 30 mol% regarding the stoichiometric amount selected (1–20 mol% Si). In this case, the incorporation of CO_3_^2−^ ions was not the cause of the significant Si uptake. The high solubility of the TMAS in the alkaline solution is likely to produce Si complex ions that are highly stable in the hydrothermal medium, giving rise to the decrease in the Si concentration in the embryo and growth steps [22,23].

A similar trend was found in the preparation of Zn-substituted HAp, where the isomorphous incorporation of Zn at the Ca site in the HAp structure was affected by the formation of Zn (OH)_x_^n+^ species, which were also stable in alkaline hydrothermal fluids at the standard pH conditions required to crystallize the HAp [20].

Although the detailed effect of the complex ion formation associated with the dopant ions in HAp has not been evaluated yet, it is important from the chemical processing point of view to enhance the control of the stoichiometry of Ca_10_(PO_4_)_6−x_(SiO_4_)_x_(OH)_2−x_ solid solutions and the particle growth at nanometer order [21,22]. In the present research work, different approaches for the synthesis of the Ca_10_(PO_4_)_6−x_(SiO_4_)_x_(OH)_2−x_ particle were investigated, devoted to investigating the chemical reaction pathway in Si^4+^-saturated solutions under hydrothermal conditions assisted by microwave heating.

The fast reaction kinetics triggered by the microwave heating in conjunction with the saturation level of Si^4+^ would achieve a broad compositional control to produce Ca_10_(PO_4_)_6−x_(SiO_4_)_x_(OH)_2−x_ compounds. The feasibility of controlling the particle size at nanometric order is likely to proceed due to Si complex ions in the hydrothermal medium, which would operate as templates in the particle crystallization process.

## 2. Materials and Methods

### 2.1. Materials

Preparation of the reagents for the synthesis of the stoichiometric pure hydroxyapatite (HAp) and silicon-substituted hydroxyapatite (Si-HAp) powders was carried out as follows; all the chemicals of reagent grade (Sigma Aldrich, St. Louis, MO, USA, 99.99% purity) were used without further purification. The 1 M Ca^2+^ and 0.2M P^5+^ stock solutions were prepared by dissolving calcium nitrate tetrahydrate (Ca(NO_3_)_2_4H_2_O) and sodium tripolyphosphate (Na_5_P_3_O_10_) in distilled water, respectively. The Si^4+^ stock solutions of three different concentrations of 0.3, 0.9, and 1.8 M were prepared by dissolving was tetramethylammonium silicate solution [(C_4_H_13_NO_5_Si_2_) (TMAS)]. Furthermore, all aqueous TMAS solutions were adjusted to pH = 10 with 7M NH_3_ solution before making up the final volume of the TMAS stocks. The 7M of NH_3_ stock solution was prepared by mixing 82.6 mL of conc. NH_3_ solution with 17.35 mL of water. 2-Propanol was added as a buffer to prevent the formation of another phosphorous species during the reaction [23,24,25].

### 2.2. Microwave-Assisted Hydrothermal Synthesis

A mother solution constituted by 17.5 mL of the 1 M Ca^2+^ solution and 15 mL of 2-propanol was magnetically stirred for 5 min. The added 2-propanol is used as a pH buffer to prevent the hydrolysis of calcium tripolyphosphate gel to orthophosphate ions. In parallel, a solution mixture (17.5 mL) containing P^5+^ and Si^4+^ ions was prepared according to the molar mixing Ca/(P + Si) ratio of 1.67. Therefore, the mixture volumes calculated by the molar ratio Ca:P:Si were 17.5:0, 16.45:1.05, 15.75:1.75, and 14.0:3.5, where the molar volumes correspond to the pure HAp and the selected silicon compositions of 6, 10, and 20 mol%, respectively. To investigate the effect of the Si saturation in the mother liquor, the molar volumes calculated were provided with the silicon solutions of 0.3, 0.9, and 1.8 M to investigate the effect of the Si saturation in the mother liquor, respectively. On mixing both solutions instantaneously, a white milky colloid formed, and the colloidal suspension was stirred constantly for 15 min. Them, pH of the colloidal suspension was adjusted to a value of 10.00 ± 0.1 by adding a 7.0 M NH_3_ aqueous solution dropwisely [1,14,20]. The suspension (50 mL) was then transferred to a double-walled, Teflon, high-pressure vessel, hermetically closed, and placed in the rotatory device of the microwave oven (MARS-5X, CEM Corp., Manasquan, NJ, USA), and was heated at 150 °C for 1 h. After the reaction, the powders were washed several times with deionized water until a neutral pH was achieved. The powder was dried using a freeze-drier (−47 °C, 3 MPa) to avoid aggregation of the particles. The chemical reaction occurs as Equation (1) to crystallize Ca_10_(PO_4_^3−^)_6−x_(SiO_4_^4−^)_x_(OH)_2−x_ nanoparticle crystallization under the proposed hydrothermal reaction assisted by microwave heating. In Equation (1), the ultimate content of (Si-O-Si)_3_O^−^ in the reaction products is equivalent to the subtraction of the Si^4+^ incorporated in the HAp and the nominal content supplied. Furthermore, OH^−^ deficiency in Ca_10_(PO_4_^3−^)_6−x_(SiO_4_^4−^)_x_(OH)_2−x_ results from the charge balance required to compensate the total negative valence of SiO_4_^4−^ groups incorporated in the HAp structure.

10Ca(NO_3_)_2(aq)_ + 2(Na_5_P_3_O_10_)_(aq)_+ y(C_4_H_13_NO_5_Si_2_)_(aq)_ + xNH_4_OH_(aq)_+ x(CH_3_)_2_CHOH →
Ca_10_(PO_4_^3−^)_6−x_(SiO_4_^4−^)_y_(OH)_2−x(s)_ + 10Na^+^ + xOH^−^_(aq)_ + xNH_4_^δ^^+^NO_3_^δ
^^−^_(aq)_ + xH_2_O +yCH_4_^δ^^+^ + (y−x)[(Si-O-Si)_3_O^−^)]_(aq)_(1)

### 2.3. Characterization

The crystalline phases of the obtained powders were determined by X-ray powder diffraction (XRD) analyses. Diffraction patterns were collected in a range 2θ from 10 to 80° at a scanning speed of 4°/min and a step size of 0.02° in a 2θ/θ scanning mode using an X-ray diffractometer Rigaku Ultima IV equipped with Cu Kα radiation (α = 1.54056 Å) operated at 40 kV and 20 mA. Furthermore, Rietveld refinement analyses of selected samples were carried out to determine the crystallite size and lattice parameters using the TOPAS 4.2 (Bruker AXS: Karlsruhe, Germany) software [19,20,21].

Fourier transform infrared (FT-IR) spectra were obtained at a wavelength range or 400–4000 cm^−1^ in the transmittance mode by FT-IR JASCO 4000 Hachioji (Tokyo, Japan) spectrometer, using palletized samples prepared with 5 mg of powder sample and 200 mg of KBr. In addition, Raman spectra analyses were observed in the range 200–4000 cm^−1^ by laze excitation at 514 nm using a Jobin Yvon Labram HR800 Raman Spectrometer (Horiba, Japan).

The content of Ca, Si, and P in the residual powders was quantitatively calculated from the inductively coupled spectrometry analyses data (ICP, ICPE-9000; Shimadzu Co., Kyoto, Japan). XPS spectra were recorded on a Kratos spectrometer (Manchester, UK) operated using an Al Ka (1486.6 eV) monochromatic X-ray source. The XPS analysis was carried out in ESCA Lab 220i-XL equipment (Shimadzu, Kyoto, Japan), at a vacuum of 2 × 10^−8^ mTorr, and a monochromatic X-ray source with aluminum anode operated at 1486.6 eV was used. The general spectra (survey) were obtained with a step energy of 117.4eV, and the analysis region was 0–1400 (eV) in link energy. Subsequently, high-resolution spectra of the C 1s, Ca 2p, P 2p, and O 1s signals were obtained for each sample. The high-resolution spectra were acquired with a step energy of 11.75 eV. Deconvolution of these spectra was performed by adjusting Gaussian curves, leaving their position and area without restriction. The FWHM value, however, remained fixed in each curve adjusted.

Morphology and particle size distribution were analyzed from the micrographs obtained by field emission scanning electron microscopy (FE-SEM JEOL 6500F JSM-7100F, Akishima, Tokyo, Japan) at 15 kV and 69 μA filament operating conditions. The image analysis was carried out using an Image-Pro^®^ Plus software (Rockville, MD, USA). In addition, crystalline features were investigated by using the high-resolution observations conducted in the transmission electron microscopy (HR-TEM, FEI-TITAN 300, Phillips, Eugene, OR, USA) operated at 300 kV.

## 3. Results and Discussion

### 3.1. Effect of Si^4+^ Saturation on the Hydrothermal Synthesis of Si-HAp

Typical XRD patterns of the residual products prepared under microwave-assisted hydrothermal conditions at 150 °C for 1 h are shown in Figure 1. This experimental set was aimed to prepare Ca_10_(PO_4_)_6−x_(SiO_4_)_x_(OH)_2−x_ solid solutions with nominal Si^4+^ content above the stoichiometric concentration of 0.2 M. Three levels of Si^4+^ saturation of 1.5, 4.5, and 9-fold were added to the hydrothermal medium employing the TMAS solutions of 0.3, 0.9, and 1.8 M, respectively. The Si^4+^ excess aimed to improve the efficiency in the incorporation of Si^4+^ ions substituting P^5+^ in the apatite hexagonal structure. In general, the diffraction patterns of the HAp and Si-HAp powders were indexed with that of the hexagonal apatite structure with space group P6_3_/m (176) (card JCPD 09-0432). The crystallization of the HAp and Si-HAp proceeded via a single-step reaction; this assumption is inferred due to the formation of secondary phases of calcium phosphate or calcium silicate; it did not take place during the hydrothermal treatment. Generally, when the lowest saturation level of Si^4+^ (1.5) was used, the Si-HAp samples intended to incorporate 6 and 10 mol% of Si did not have marked differences of peak intensity and sharpness from those of the pure HAp powders. The sample prepared with the 0.3 M TMAS solution with the volume to supply 0.33 mol% Si exhibited a slight shifting of the peak to a lower diffraction angle (Figure 1a). In contrast, at 4.5 and 9-fold saturation levels, a progressive displacement of the XRD pattern proceeded at small 2θ angles, and also a remarkable peak broadening occurred on the diffraction patterns, as shown in Figure 1b,c. This behavior was markedly evident in the Si-HAp samples prepared with amounts of 5.0 and 12.16 mol% Si with TMAS solutions of 0.9 and 1.8 M. Hence, the Si^4+^ excess in the aqueous phase plays an important role in promoting the uptake of Si during the crystallization of the single-phase Si-HAp powders. These crystalline aspects were not clearly analyzed for the Si-HAp prepared even in hydrothermal conditions elsewhere [14,22,26], which do not bear crystalline structural evidence, indicating that the Si content reported is bulkily incorporated inside the particle rather than near the particle surface.

### 3.2. Crystalline Structural and Chemical Compositional Analyses of Si-HAp Powders

The Rietveld refinement of Si-HA powders conducted with the refinement algorithm includes the Si^4+^ and P^5+^ molar contents determined by wet chemical analyses (Table 1, ICP results), together with the atom occupation, as suggested elsewhere (Figure 2) [1,2,5,8]. Various crystalline structural features were included as the refinement parameters, such as background, lattice parameters, scale factor, profile half width, crystallite size, local strain, thermal isotropic vectors, and spatial coordinates. The parameters selected provided high accuracy for calculating the structural features of the Si-HAp powders. The Rietveld refinement approach provided the lowest values of the goodness-of-fit factor (GOF/χ^2^), averaging 1.23 ± 0.6, and low *R*_wp_ 7.32 ± 1.0 values were obtained (Table 1), which confirm the fine structure of all Si-HAp with excellent accuracy. The calculated diffraction profiles are in good agreement with the experimental ones due to the small residual difference between the observed and calculated patterns depicted by the residual straight line, and the vertical lines correspond to the calculated Bragg peaks positions (Figure 2) [8]. The calculated lattice parameters for pure HAp and Si-HAp are given in Table 1. Both the *a*_o_ and *c*_o_ axes increase slightly with the Si^4+^ incorporation into the pure HAp powders; whilst the length of both the *a*_o_ axis (9.4450 to 9.4365 Å) and the *c*_o_ axis (6.8882 to 6.8824 Å) decreased as the Si^4+^ incorporation ratio increased in Si-HAp powders, leading to the decrease in cell volume, specially shown in the Si-HAp powders prepared at a high Si^4+^ saturation level. It should be mentioned that in the case of Si-HAp powders at the maximum Si^4+^ incorporation of 12.16 mol%, the least “*a*_0_” and “*c*_0_” lattice parameters were obtained in comparison with those powders containing below 10 mol% Si contents. These were caused by proportional release of OH^−^ ions, which compensate the negative charges provided by SiO_4_^4−^ ions substituting PO_4_^3−^ in tetrahedral positions, located parallel to the *c*-axis along the tunnels located at the HAp structure honeycomb edges. This phenomenon provokes a bulk shrinkage of the hexagonal unit cell volume (530.75 Å^3^) in the HAp structure, which would support the inference that Si is bulkily incorporated in the Si-HAp during the crystallization process, achieved by fast reaction kinetics triggered by the microwave heating under hydrothermal conditions. The maximum content of Si^4+^ (12.16 mol%) incorporated in HAp in the current experiment is above that reached under conventional hydrothermal [22] and coprecipitation [4] conditions.

FT-IR spectra of the HAp and Si-HAp powders revealed hydroxyl (OH^−^) groups stretching (3571 cm^−1^) and vibration (631 cm^−1^) bands (Figure 3). Furthermore, the strong bands at 1086, 1014, and 960 cm^−1^ wavenumbers are associated with the stretching vibration modes of PO_4_^3−^ tetrahedral group. The doublet band between 593 and 572 cm^−1^ corresponds to O-P-O bond bending mode and these results agree with the data reported elsewhere [14]. The shoulder peaked at 897 cm^−1^, and this band is assigned to the Si-O-Si vibration mode for tetrahedral SiO_4_^4−^ groups. A marked distortion of the shoulder peak together with the signals *v*_1_ and *v*_3_ of the P-O-P and PO_4_^3−^ major bands was found with progressive increases in the SiO_4_^4−^ molar content in the Si-HAp powders, as reported previously [24,26]. Furthermore, *v*_1_ and *v*_3_ bending modes of the P-O-P band gradually decreased in their absorbance by increasing the uptake of Si in the HAp structure, and a slight displacement to lower wavenumbers was revealed on the Si-HAp samples (Figure 3). In comparison with the FT-IR spectrum of pure HAP, the peak at 1086 cm^−1^ (PO_4_^3−^) in Si-HAp samples decreased as the silicon content uptake increased, due to the structural change in the HAp lattice [24,25,26,27]. Additionally, the intensity of the OH^−^ group band on the 0.16 and 0.90 mol% Si samples was similar irrespective of the saturation contents of Si^4+^. On the contrary, the OH^−^ stretching symmetrical band at 3571 cm^−1^ and bending OH^−^ 631cm^−1^ markedly deceased in its absorbance in Si-HAp powders prepared with the highest concentration 1.8 M of TMAS (Figure 3c). XRD signals and FT-IT spectra confirm that the bulk Si should be incorporated in the apatite structure rather than partially existing at the particles’ surface [26].

These structural results are consistent with the substitution mechanism proposed, where PO_4_^3−^ ions are replaced by SiO_4_^4−^ ions, causing a stoichiometric release of OH^−^ ions, which maintains the total charge balance in the HAp structure (Equation (1)). Under hydrothermal conditions, the saturation of Si^4+^ probably induced a reduction in the amount of hydroxyl groups to compensate for an extra negative electric charge produced by the incorporation of the silicate groups, and the formation of OH^−^ vacancies (*V*) might have taken place to maintain the charge balance neutrality, as is described by the following equation: PO_4_^3−^ + OH^−^ → SiO_4_^4−^ + *V*_(OH)−_. Indeed, this behavior is consistent with previous research work [13].

Raman analyses were carried out to determine detailed crystalline differences in the bonds. Raman spectra of the SiHAp constituents in Figure 4a show that the *v_1_* symmetric stretching PO_4_ mode at around 996 cm^−1^ corresponds to the HAp structure of the prepared Si-HAp powders samples with both concentrations (0.9 and 1.8 M). Other typical PO_4_ peak modes of the bending *v*_2_, asymmetric stretching *v*_3_, and bending *v*_4_ (PO_4_) were also observed at around 400, 1100, and 600 cm^−1^, respectively. The progressive decrease in the peak intensity and their broadening for all PO_4_ *v*_2_–*v*_4_ bands confirmed the SiO_4_^4−^substitution by PO_4_^3−^ group. The OH^−^ peak intensity decreased with broadening of the band [20]; these results are in line with those of FT-IR and XRD. Under hydrothermal conditions assisted by microwave heating, the molar percentage of Si^4+^ substitution was larger than 10 mol% by providing a 9-fold saturated Si^4+^ precursor in comparison with the stoichiometric limit of acceptance to keep the stability of the HAp structure. This explains the low variation of the lattice parameter and very small variation in the crystallite size, because only 0.33 mol% of Si^4+^ was incorporated in the HAp structure for samples prepared with a low concentration of TMAS (0.3 M). However, Si-HAp samples synthesized with a greater content of Si^4+^ (12.16 mol%) did not show the presence of the characteristic Si signal, and only small changes were detected in the vibration OH^−^ signal at 3570 cm^−1^, which were slightly decreased and broadened under the high Si^4+^ saturation of TMAS (1.8 M) during the hydrothermal reaction.

Figure 5a–d shows the XPS spectra for HAp and Si-HAp powders prepared with a 0.9 M solution for samples with ultimate contents of 5 and 12.16 mol% Si. The XPS spectra corresponding to the photoelectron core levels of Ca 2p, P 2p, Si 2p, and O 1s without any additional core level of other elements were detected by the XPS analyses in the survey spectrum. Generally, the Ca 2p spectrum recorded in the binding energy (BE) range of 344.0–355.0 eV revealed the doublet associated with Ca-O bonds in the HAp and Si-HAp samples, which are constituted by the core level Ca 2p_3/2_ at 347.21 eV and Ca 2p_1/*3*_ at 350.55 eV BE. Furthermore, the P 2p peak is symmetric and its average BE is at 132.9 eV both for HAp and Si-HAp (Figure 5b). Moreover, the O 1s core level peak was deconvoluted into two peaks, and a small peak, which fits the shoulder between 532 and 534 eV, was detected. The deconvolution indicates that the peak average BE energy was of 532.45 eV, and this peak is associated with the SiO_4_ units in the prepared powder samples, where the gradual increase in peak intensity in the samples synthesized with different Si^4+^ contents supports our inference (Figure 5c). The second large peak, deconvoluted at an average BE of 530.8 eV, corresponded to the O 1s core level of PO_4_^3−^ tetrahedral units. However, the presence of silicon was very clear in powders prepared with 0.41 mol% Si, as shown in Figure 5d. In contrast, a symmetric peak corresponding to the core level Si 2p at BE of 103.3 eV was revealed in the Si-HAp samples obtained with the molar volume corresponding to 5.0 mol% Si using the TMAS solution of 0.9 M.

The XPS spectra in Figure 6 indicated that HAp and Si-HAp powders prepared in the presence of the highest Si^4+^ saturated TMAS solution (1.8 M) exhibited a gradual uptake of Si. All the samples are constituted by the chemical elements that form the HAp structure. Figure 6a shows the typical doublet peak associated with the core level Ca 2p_3/2_ XPS BE at 347.1 eV and Ca 2p_1/3_ at 350.07 eV. The doublet peaks increased slightly as the silicon incorporation content increased in the synthesized samples obtained with 1.61 mol% of Si in the Si-HAp powders. Likewise, the symmetric signals of P 2p increased slightly with the increase of Si^4+^ content, achieving a binding energy of 132.92 eV. Furthermore, the binding energy signals for O 2p peaks were nearly symmetric (Figure 6c). The main component at 530.9 eV corresponding to the O^2−^ is linked only to a phosphorus atom as in PO_4_^3−^ ions of the HAp structure. With the increase of the mol% of SiO_4_^4−^ in the prepared powders, the BE signal shifted slightly into 530.79 eV and a shoulder peak at an average BE of 532. 53 eV increased [11]. Therefore, this shoulder peak is attributed to the O^2−^ ions associated with the SiO_4_^4−^ tetrahedral units.

In addition, the presence of Si was confirmed in the HAp structure of the powders prepared using the highest saturation of Si (9-fold, Figure 6d). The XPS core level Si 2p corresponded to the symmetric peak at BE of 103.18 eV for the 1.61 mol% Si. The Si 2p peak intensity increased as a result of the improved SiO_4_^4−^ incorporation in the HAp structure. The largest peak at a BE of 103.31 eV occurred in the sample containing 12.16 mol% Si, and this SiHAp powder exhibited similar BE behavior as was reported elsewhere [20,21]. The lack of Si uptake in the HAp structure would be caused by the highly soluble species produced by the TMAS precursor, such as (Si-O-Si)_3_O^−^. This anionic specie is preferentially formed in methanolic solutions due to the high solubility property of TMAS [22,23,24,25,26,27]. Therefore, these species should reduce the solute concentration of SiO_4_^4−^ ions at the supersaturation stage reached under microwave-assisted hydrothermal conditions. This behavior is attributed to the fact that the TMAS might dissolve in 2-propanol under the hydrothermal fluid forming silicate ions in the medium. The highly soluble silicate ions themselves promote polymerization of silicate ions, limiting the amount of OH^−^ ions in the SiHAp powders, because the related vibration OH^−^ signal in FT-IR spectra decreased slightly under the high Si^4+^ saturation of TMAS (1.8 M) during the hydrothermal reaction.

### 3.3. Morphological Aspects of the Partially Substituted Si-HAp Particles Prepared Hydrothermally

The morphology of HAp and Si-HAp powders was analyzed by FE-SEM micrographs (Figure 7). Pure HAp was determined to be monodispersed particles with a regular rod-like morphology with an average size of 62 nm. In contrast, the Si-HAP powders were mostly monodispersed Si-HAp agglomerates with a quasi-oval shape. The agglomerates average size of the Si-HAp samples incorporating 0.9 mol% of 5.0 mol% Si contents was between 235.5 ± 29.7and 297.4 ± 19.4 nm. Whilst, when the 9-fold Si^4+^ sutured TMAS solution (1.8 M) was used, the agglomerated average size of the samples with 6.11 mol% and 12.16 mol% Si contents was between 243.9 ± 22.2 and 315.1 ± 22.5 nm. The excess of Si in the hydrothermal medium should trigger the formation of Si-HAp agglomerates prepared under fast kinetic reaction conditions assisted by the microwave heating. These results are supported by variation in the crystallite size, as calculated in the Rietveld refinement results (Table 1).

In addition, the detailed crystalline structure features of the quasi-oval agglomerates of Si-HAp powders containing 1.62 mol% Si and 12.16 mol% Si, prepared in the presence of the 1.8 M TMAS solution, were investigated by HR-TEM observations (Figure 8a,d). These images revealed that the bulk morphology of the quasi-oval shaped agglomerates is irrespective of the Si^4+^ saturation in the reaction fluid medium. Generally, the agglomerates are formed by nanosized euhedral rod-shaped crystals with varied sizes. The rod-shaped crystals containing 1.62 mol% Si exhibited a broad monomodal length distribution, and the average length was 32 ± 8.0 nm (Figure 8a,b). On the contrary, a slight reduction in the crystal length occurred in the Si-HAp sample incorporating 12.15 mol% Si, which was shorter than in the sample incorporating 1.62 mol Si, and the length size distribution curve revealed it to be in a vast proportion (27.0 ± 8.0 nm) (Figure 8d,e). The Si- HAp particle size gradually decreased with the increase of Si content. In addition, HR-TEM and SAED provide a high crystallinity of the euhedral rod-shaped SiHAp particles incorporating both 1.62 mol% and 12.15 mol% Si (Figure 8c,f). The SAED pattern (inset) of the squared area in Figure 8c indicates that the preferential stacking of the 1.62 mol% Si-HAp crystals proceeds along the hexagonal structure basal plane with a Miller index of (300), although an irregular atomic stacking occurred in some areas of the agglomerate 12.15 mol% Si. Meanwhile, the SAED pattern of the crystals indicated that the agglomerates containing less silicon were associated with the family plane <112> Miller index. The interplanar spacing calculated for the (112) and (211) planes was 0.32 nm and 0.27 nm, respectively. These values are very close to the interplanar spacing positions in the single-phase hydroxyapatite structure. The SAED patterns confirmed that the fine, rod-like Si-HAp crystals are single crystals.

SEM and HR-TEM observations indicated that the Si^4+^ excess in the hydrothermal medium led to a marked variation in the growth and the spontaneous assembly process of the euhedral rod-like particles. These differences should be generated from a different dissolution-crystallization mechanism, which achieved rapid reaction kinetics by the microwave heating of the hydrothermal medium at 150 °C. However, the formation of highly soluble anionic silicate tri-branching units (Si-O-Si)_3_O^−^ (Q^3^) could form to become dominant under the current hydrothermal conditions due to the addition of 2-propanol [28]. The Q^3^ units, which can act as fine micelles that trap the Si-HAp nutrients, led to a certain supersaturation state in the hydrothermal reaction conditions, which should be essential for the crystallization of the irregular oval-shaped Si-HAp agglomerates. When a further excess of Si^4+^ was supplied, the molar volume of the Q^3^ units increased, leading to an increase in the agglomerate size. The above inference, associated with the reaction pathway, is supported by the fact that no crystalline SiO_2_ species were formed as a secondary phase as a result of the crystallization of the Q^3^ units. Hence, we surmise that the hydrothermal microwave-assisted method is efficient to produce Si-HAp powders with larger contents of silicon rather than those techniques reported recently [12] including the conventional hydrothermal process [22]. This method coupled with the use of TMAS has the potential for processing Si-HAp to prepare biomaterials with implant applications in medicine.

## 4. Conclusions

Si-HAp powders were successfully crystallized under fast microwave-assisted hydrothermal synthesis conditions at a low temperature (150 °C) for 1 h using saturated Si^4+^ precursor TMAS solutions.

The maximum amount of Si^4+^ incorporated in the HAp structure was 12.16 mol%, using an excess of 1.8 M of TMAS. The addition of highly concentrated Si^4+^ solutions (0.3–1.8 M) caused differences on the crystalline unit cell of the apatite and produced agglomerates constituted by fine euhedral rod-like crystal with an average length of 27.0 ± 8.0 nm and a single crystal habit. The Si^4+^ excess in the reaction media led to the rod-like crystal self-assembly to produce irregular oval-shaped Si-HAp agglomerates, which were prepared under fast kinetic reaction conditions assisted by the microwave heating and exhibit sizes between 233.5 and 315.1 nm. These agglomerates exhibited a marked size coarsening, which was triggered by the Si^4+^ saturation level supplied in the hydrothermal media. Despite the Si^4+^ ion high level of saturation used in the hydrothermal reaction medium, the lack of Si incorporation in the HAp structure is promoted by the Q_3_ species, namely (Si-O-Si)_3_O^−^, which are likely formed in hydrothermal media containing 2-propanol. These highly soluble ions reduced the supersaturation SiO_4_^4−^ molar volume in the medium, almost 50% below the ultimate stoichiometric content selected. Furthermore, a remarkable decrease in the O− ions content was confirmed by FTIR and XPS analyses, and the gradual OH^−^ lost was caused to compensate for the partial incorporation of SiO_4_^4−^ at tetrahedral PO_4_^3−^ sites in the HAp structure. The present hydrothermal microwave-assisted method has delivered high processing efficiency to crystallize Si-HAp particles with a control on the Si^4+^ content. This method has potential for processing Si-HAp bioceramic implants in medicine.

## Figures and Tables

**Figure 1 nanomaterials-11-01548-f001:**
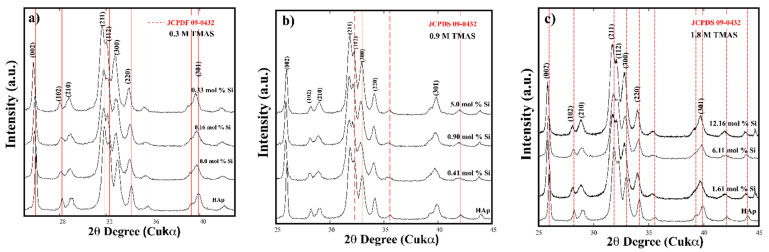
Si-HAp XRD patterns of samples synthesized under microwave-assisted hydrothermal conditions at 150 °C for 1 h, pH = 10, using 0.2 M Na_5_P_3_O_10_ with different Si^4+^ mol% using (TMAS), which was mixed with a molar excess of (**a**) 1.5 (0.3 M), (**b**) 4.5 (0.9 M), and (**c**) 9 (1.8 M) folds, above the stoichiometric concentration of 0.2 M as shown.

**Figure 2 nanomaterials-11-01548-f002:**
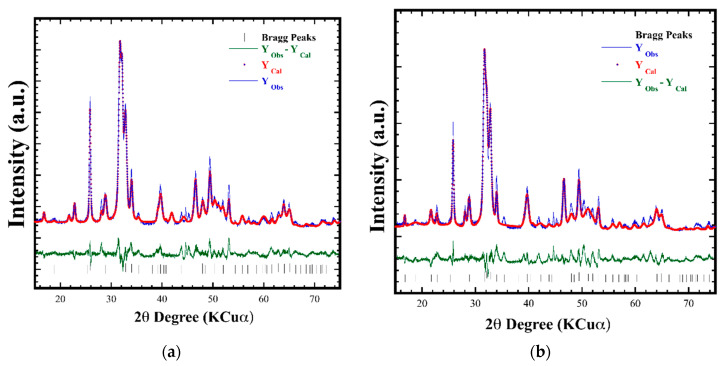
Rietveld refinement plot of Si-HAp powders prepared with different content of Si^4+^: (**a**) 6 mol% and (**b**) 20 mol% using (TMAS) [(CH3)_4_N(OH).2SiO_2_] with a molar ratio of 1:3.0 (1.8 M), precursor, under microwave-assisted hydrothermal conditions at 150 °C for 1h, pH = 10.

**Figure 3 nanomaterials-11-01548-f003:**
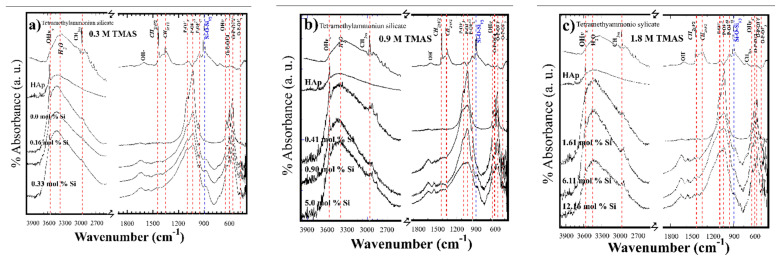
Fourier transform infrared (FT-IR) spectrum analysis of the Si-HAp powders obtained by microwave-assisted hydrothermal process at 150 °C for 1 h using different concentrations of [(C_4_H_13_NO_5_Si_2_) (TMAS)]: (**a**) 0.3 M; (**b**) 0.9 M, and (**c**) 1.8 M.

**Figure 4 nanomaterials-11-01548-f004:**
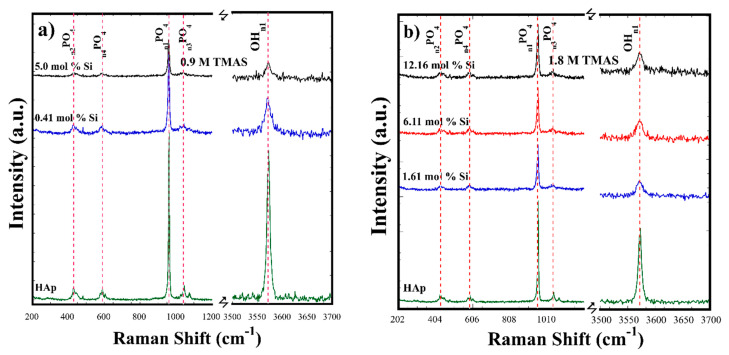
Raman spectra of the Si-HAp powders obtained by microwave assisted hydrothermal process at 150 °C for 1 h using different molar concentrations of the TMAS solutions: (**a**) 0.9 and (**b**) 1.8 M.

**Figure 5 nanomaterials-11-01548-f005:**
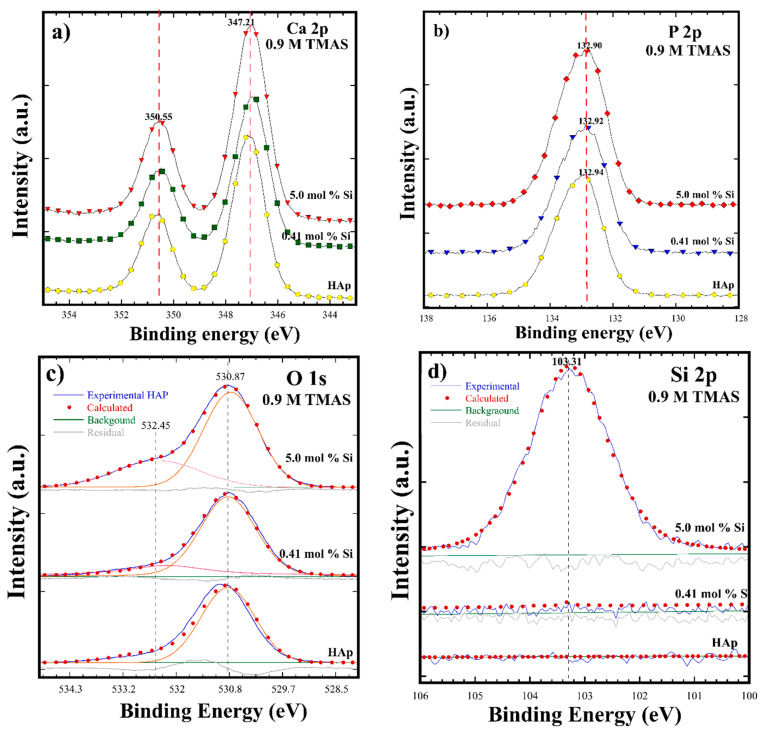
The XPS analysis for HAp and Si-HAp powders obtained by microwave assisted hydrothermal process at 150 °C for 1 h using 0.9 M of TMAS and different mol% of Si: (**a**) Ca 2p; (**b**) P 2p; (**c**) O 1s; (**d**) Si 2p.

**Figure 6 nanomaterials-11-01548-f006:**
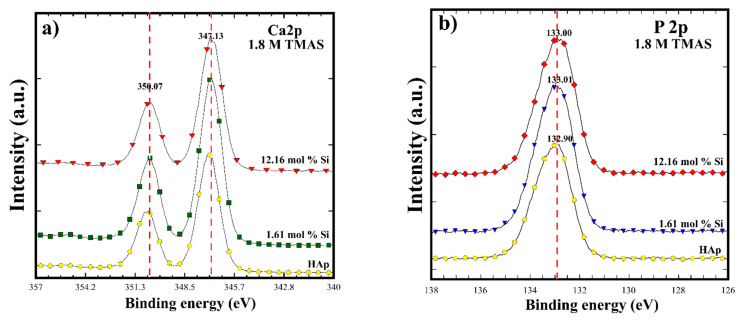
The XPS analysis for HAp and Si-HAp powders obtained by microwave assisted hydrothermal process at 150 °C for 1 h using 1.8 M of the TMAS solutions with different mol% Si: (**a**) Ca 2p; (**b**) P 2p; (**c**) O 1s; (**d**) Si 2p.

**Figure 7 nanomaterials-11-01548-f007:**
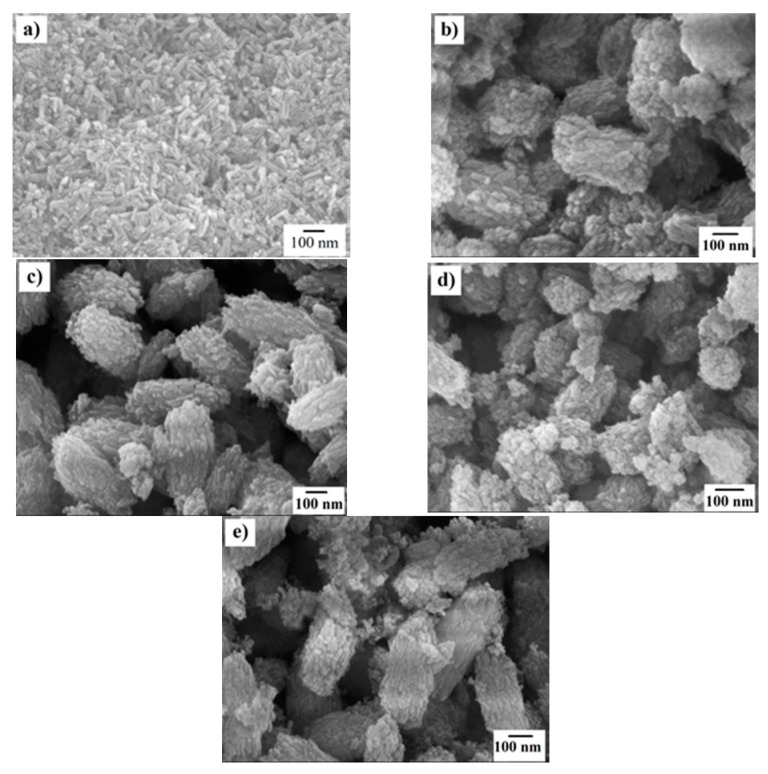
FE-SEM microphotographs of HAp and Si-HAp powder prepared by microwave-assisted hydrothermal process at 150 °C for 1 h, (**a**) 0 mol%; and using Si saturated [(C_4_H_13_NO_5_Si_2_) (TMAS)] with different nominal concentrations of TMAS 0.9 M and (**b**,**c**) and 1.8 M (**d**,**e**) and different mol% of Si: (**b**,**c**)10 mol% and (**d**,**e**) 20 mol%, respectively.

**Figure 8 nanomaterials-11-01548-f008:**
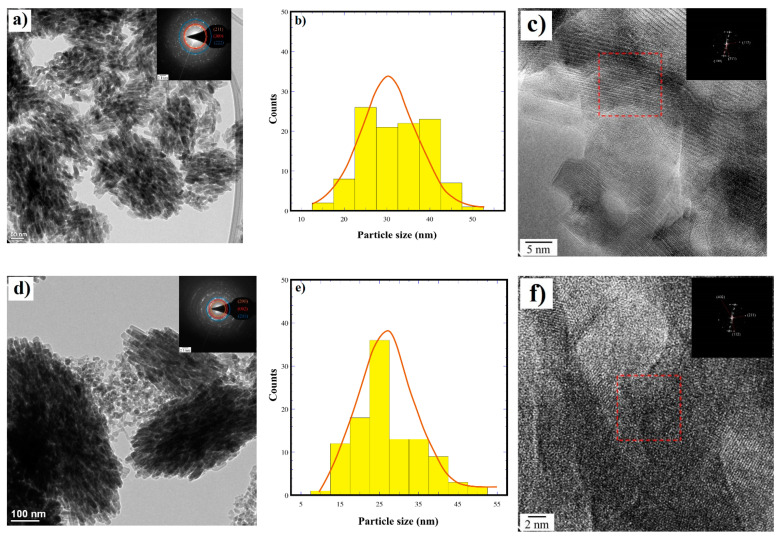
HRTEM micrographs of the Si-HAp powders prepared under an assisted hydrothermal process at 150 °C for 1 h using [(C_4_H_13_NO_5_Si_2_) (TMAS)] 1.8 M: (**a**–**c**) 1.62 mol% of Si and (**d**–**f**) 12.15 mol% of Si.

**Table 1 nanomaterials-11-01548-t001:** Chemical and physical features of Si-HAp powders prepared under hydrothermal-assisted microwave treatments at 150 °C for 1 h using different concentrations of TMAS solutions and tripolyphosphate as a P precursor.

Sample * Id Reference	Molar Concentration (±0.001) of Si^4+^	Nominal Mol %		Chemical Compositon ^a^	Molar Ratio Ca/P	GOF χ^2 b^	Lattice Parameter ^b^			Strain ^b^	R_wp_	R_Bragg_	Crystallite Size (nm)
		PO_4_	SiO_4_				*a*_0_ (Å)	*c*_0_ (Å)	Cell Volume (Å^3^)				
HAp		100	0	Ca_10_(PO_4_)_6_(OH)_2_	1.667	1.36	9.4248(7)	6.8764(5)	528.97(0.09)	0.5 (0.02)	9.43	1.44	51.69 (2.3)
MM62	0.3	94	6	Ca_10_(PO_4_)_6_(SiO_4_)_0.0_(OH)_2_	1.667	1.28	9.4433(10)	6.8886 (7)	532.00(0.12)	0.68 (0.03)	8.58	1.63	46.14 (3.4)
MM66	0.3	90	10	Ca_10_(PO_4_)_5.99_(SiO_4_)_0.01_(OH)_1.99_	1.667	1.66	9.4447(18)	6.8869 (8)	532.10(0.22)	0.53 (0.08)	10.89	3.08	46.64 (4.9)
MM63	0.3	80	20	Ca_10_(PO_4_)_5.98_(SiO_4_)_0.02_(OH)_1.99_	1.667	1.74	9.4439(16)	6.8872(7)	531.89(0.12)	0.38(0.04)	11.15	3.93	46.18(2.9)
MM50	0.9	94	6	Ca_10_(PO_4_)_5.975_(SiO_4_)_0.025_(OH)_1.975_	1.674	1.64	9.4443(13)	6.8866(11)	531.97(0.19)	0.56(0.07)	10.86	3.18	45.03(3.7)
MM51	0.9	90	10	Ca_10_(PO_4_)_5.946_(SiO_4_)_0.054_(OH)_1.946_	1.682	1.77	9.444413)	6.8866(10)	532.00(0.17)	031 (0.05)	11.54	4.11	36.96(2.1)
MM53	0.9	80	20	Ca_10_(PO_4_)_5.699_(SiO_4_)_0.301_(OH)_1.699_	1.755	1.27	9.4424(9)	6.8863(6)	531.73(0.11)	0.30(0.03)	8.24	1.71	35.94(1.2)
MM64	1.8	94	6	Ca_10_(PO_4_)_5.908_(SiO_4_)_0.097_(OH)_1.908_	1.692	1.64	9.4475(16)	6.8884(11)	532.46(0.20)	0.61(0.06)	10.56	2.95	48.82(5.1)
MM68	1.8	90	10	Ca_10_(PO_4_)_5.633_(SiO_4_)_0.367_(OH)_1.633_	1.775	1.30	9.4447(11)	6.8872(8)	532.09(0.14)	0.68(0.04)	8.44	1.81	48.64(3.4)
MM65	1.8	80	20	Ca_10_(PO_4_)_5.271_(SiO_4_)_0.729_(OH)_1.271_	1.897	1.11	9.4365(9)	6.8824(7)	530.75(0.11)	1.68(0.09)	2.63	1.15	39.87(3.6)

^a^ Chemical formula of Si-HAp powders was calculated from Ca, P, and Si contents determined via ICP-AES analysis, and OH^−^ was calculated by charge balance. ^b^ Values obtained from the refinement of XRD patterns carried out using the Rietveld method. (* Identification number samples: MM62, MM66, MM63; MM50, MM51, MM53; MM64, MM68 and MM65 correspond to the SI-HAp with 5, 10 and 20 mol% with concentration 0.3, 0.9 and 1.8 M respectively).

## Data Availability

The data presented in this study are available in the Hydrothermal Synthesis of Nanoparticles knowledgebase https://www.mdpi.com/journal/nanomaterials/special_issues/hydrotherm_nano, accessed on 10 May 2021.

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
