# Peer review of "Preparation of Silicon Hydroxyapatite Nanopowders under Microwave-Assisted Hydrothermal Method"

_nanomaterials, 2021, doi:10.3390/nano11061548_

Round 1

Reviewer 1 Report

The topic is interesting, the data presented are applicable, and the experiments were properly performed, but passably presented / discussed. The experimental work presented is original and may be interesting. However, this study suffers due to the absence of a clearly presentation of the up-to-date view of the most important/ recent progress in the field (Introduction) and lack of proper discussion (Results and discussion). Furthermore, the written is very week. In its current state, the level of English throughout the manuscript needs language improving. Please check the manuscript and improve the language carefully.

Introduction. The Introduction hints the reader from a general subject area to a particular topic of inquiry, giving some background information and set the scope, context and significance of the research. The authors should improve the literature review in order to present recent, abundant state of the art in the field of manuscript, while keeping only the relevant parts.

Results and discussion. The interpretation of the obtained results/correlations is not detailed/ performed in relationship with past (recent, valuable) studies on the topic.

I consider that the article can be accepted for publication only after a major, major revision.

Author Response

About the reviewer comments in terms of “ this study suffers due to the absence of a clearly presentation of the up-to-date view of the most important/ recent progress in the field (Introduction) and lack of proper discussion (Results and discussion)

Thank you for the time and comments that are relevant to improve the manuscript technical quality.

1- Introduction. The Introduction hints the reader from a general subject area to a particular topic of inquiry, giving some background information and set the scope, context and significance of the research. The authors should improve the literature review in order to present recent, abundant state of the art in the field of manuscript, while keeping only the relevant parts.

First of all, thank you for your time, your comments and kind suggestions that are relevant to improve the manuscript technical quality.

However, I do not understand your point of view. Because, to support the information on the introduction section for state of the art in the present manuscript, we did a review of scientific papers that had a close relationship with the synthesis of Si-HAp powders. Also, we did a deep literature review of the data base publications relate with the “Synthesis of Substituted hydroxyapatite”, considering different methodologies and methods, however we did not found such huge number of papers directly related with the present manuscript.  On the other hand, certainly there are many research orientate to the study the Invitro biocompatibility of the Si-HAp

¿Could you please let me know, which particular paper It will necessary to include?? Could you please be more specific?.

Furthermore, I would like to point that: In the introduction section of the presented manuscript we took in a count the most important and relevant literature review, that was most related to our research. Also we did make a deep review of the data base publications relate with the Synthesis of Substituted hydroxyapatite, we did not found such huge number of papers directly related with the present manuscript.

In addition, in the present manuscript  you can found in highlight the relevance of the present manuscript , that can be found on pages 2-3 lines between 90 to 101, where we are emphasizing that on previous researches did not study directly the influence of the excess of Si4+ precursor and its effect during substitution to obtain Si-HAP under hydrothermal microwave assisted synthesis, nether any other technique.

2- (Results and discussion). Results and discussion. The interpretation of the obtained results/correlations is not detailed/ performed in relationship with past (recent, valuable) studies on the topic.

I want to apologize, because I do not understand clearly what do you mean?

In this case, throughout the Results and Discussion Section of the  present manuscript ; We had discuss and explain the improvements of our results in comparison with the previous reported research.

In addition, the relevance of the present research is relative to obtained the Si-HAP nanopowders with high level of substitution (12.16 mol%).This level of substitution of Si4+ was higher than that it was previously reported. Furthermore, it was explain the effect of supersaturation of Si4+  used as precursor (TMAS) during the substitution of the PO43- by SiO 4-. From Page 12 and line 434 to 453.

  1. In its current state, the level of English throughout the manuscript needs language improving. Please check the manuscript and improve the language carefully.

Thank very much for your time and comments and for your time:

We apologise for the inconvenience caused by this matter. The manuscript carefully was reviewed for the English grammar usage by the English Proofread by PRS professional native English check, previously, to the submission, also, the certificate was attached to the cover letter at the beginning.  

Thank very much for your time and all the suggestion that were consider valuable to improve the manuscript

Reviewer 2 Report

Manuscript "Preparation of silicon hydroxyapatite nanopowders under microwave-assisted hydrothermal method" by Z. Matamoros-Veloza, JC Rendon-Angeles, K. Yanagisawa, T. Ueda, K. Zhu, B. Moreno Perez presents a method for the synthesis of silicon dopped hydroxyapatite and characteristics of the obtained samples by means of: XRD, FTIR, Raman and XPS spectroscopy, Fe-SEM and HR-TEM microscopy.

It has been shown that silicon is essential for metabolic processes and biomineralization. Adding hydroxyapatite to silica improves its bioavailability properties in living organisms. Research on silicon-substituted hydroxyapatite intensified after 2005, and since then several dozen, about 30, papers are published annually on this subject.

The novelty of the reviewed work is the synthesis of silicon-substituted hydroxyapatite using the sol-gel method supported by microwave radiation. The methodology of testing hydroskyapatite samples used in the work is correct.

Retvield refinment was used to analyze the diffraction patterns. However, the crystallite sizes obtained with this method, Table 1, indicate that an isotropic microstructure was assumed in the optimization. Due to the hexagonal structure of hydroxyapatite, the hydroxyapatite crystals have the shape of a hexagonal polyhedron, this is shown in Fig. 8 a and d. In such cases, the crystallite size is calculated by the Schrerr method for reflections 002 and (300) e.g. G.C. Koumoulidis et al., J. Colloid Interface Sci. 259 (2003) 254-260.

Carbonates and carbon dioxide show a high adsorption affinity to hydroxyapatite, because during the synthesis it is not always possible to avoid contamination of hydroxyapatite samples with carbonate ions, it is advisable to pay attention to this problem. In the presented study did not analyze the presence of carbonates in hydroxyapatite samples. The FTIR and Raman spectra shown are too small in size to conclude whether the samples contained carbonates or not. I believe that a careful analysis of the FTIR, Raman and XPS spectra in the C1s region can explain this problem.

Line 153 is TOPAZ 4.2 shouldn't it be TOPAS 4.2?

Summarizing, I believe that the presented manuscript requires corrections before publication in Nanomaterials.

Author Response

  1. Response to Reviewer 2

Manuscript "Preparation of silicon hydroxyapatite nanopowders under microwave-assisted hydrothermal method" by Z. Matamoros-Veloza, JC Rendon-Angeles, K. Yanagisawa, T. Ueda, K. Zhu, B. Moreno Perez presents a method for the synthesis of silicon dopped hydroxyapatite and characteristics of the obtained samples by means of: XRD, FTIR, Raman and XPS spectroscopy, Fe-SEM and HR-TEM microscopy.

Thank you for the time and comments that are relevant to improve the manuscript technical quality.

  1. It has been shown that silicon is essential for metabolic processes and biomineralization. Adding hydroxyapatite to silica improves its bioavailability properties in living organisms. Research on silicon-substituted hydroxyapatite intensified after 2005, and since then several dozen, about 30, papers are published annually on this subject.

Thank you for your detailed evaluation and comments.

In the present manuscript we already consider enough number of more relevant paper that are more related to our research. Of course we can include additional references If your consider will be necessary, so ¿Could you please let us to know exactly which reference(s) will contribute to clarify the present manuscript?.

  1. The novelty of the reviewed work is the synthesis of silicon-substituted hydroxyapatite using the sol-gel method supported by microwave radiation. The methodology of testing hydroskyapatite samples used in the work is correct.

Thank you for the time and comments that are relevant to improve the manuscript technical quality.

In the present technique, we call coprecipitation and hydrothermal treatment assisted by microwave heating using specific autoclaves for this treatment using  double-walled Teflon high pressure vessel and placed in the rotatory device of the microwave oven (MARS-5X, CEM Corp., USA), hermetically closed and was heated at 150 °C for 1 h, as it was mentioned on materials and method section. 

  1. Retvield refinment was used to analyze the diffraction patterns. However, the crystallite sizes obtained with this method, Table 1, indicate that an isotropic microstructure was assumed in the optimization. Due to the hexagonal structure of hydroxyapatite, the hydroxyapatite crystals have the shape of a hexagonal polyhedron, this is shown in Fig. 8 a and d. In such cases, the crystallite size is calculated by the Schrerr method for reflections 002 and (300) e.g. G.C. Koumoulidis et al., J. Colloid Interface Sci. 259 (2003) 254-260.

Thank you for the time and for the comments again; I want to point out that by using Rietveld refinement, the software itself in including the crystallite size, and  normally with the rietved analysis it is possible to achieve greater accuracy of the reflections

The main crystallographic data conforms a subroutine of the program algorithm designed to conduct the structural refinement. The algorithm is based in a 10-coefficient shifted Chebyshev polynomial function for modelling the background, and a pseudo-Voigt function fitted the profile peak shape. The refinement approach calculates the unit lattice cell parameters, the isotropic thermal displacement, the crystallite size and each secondary phase content. Rietveld refinement algorithm calculated the content of each phase identified, and the schematic results are shown in table 1.

In addition, the crystallite size is calculated by the Schrerr method it’s used when the Rietveld refinement  pattern it not included. Anyway, we did calculate using the Schrerr method for example for the simple prepared with the highest Si4+ content (12.16 mol% and 1.8M) , and the XC= 28.0±1.0 nm, this value its lowest than that obtained by Rietveld refinement; 39.87 nm.

  1. Carbonates and carbon dioxide show a high adsorption affinity to hydroxyapatite, because during the synthesis it is not always possible to avoid contamination of hydroxyapatite samples with carbonate ions, it is advisable to pay attention to this problem. In the presented study did not analyze the presence of carbonates in hydroxyapatite samples. The FTIR and Raman spectra shown are too small in size to conclude whether the samples contained carbonates or not. I believe that a careful analysis of the FTIR, Raman and XPS spectra in the C1s region can explain this problem.

Thank you again for your kind suggestion and comments:

Certainly, the carbonates  and CO2 can adsorbed into the HAp, however in this particular case, there are several  factors that we did not included the results on the manuscript;  because as you had mentioned CO3 or CO2 can absorbed  during the characterization ( FT-IR, Raman, or XPS) . This reason was that  it  was not relevant  to included because if  the detected CO3  or CO2  during the  synthesis, could be affect the XRD patterns, however  we did not identify any peak relate to those cations.  Indeed we did consider  XPS  spectra, for C1s region, but we did not  include on the present results. ( became to long the paper).   However if you think will be necessary to include, we can do it.

Anyway  the C1s signals were assigned to the absorbed CO2,during the analysis. and no great changes were observed on the samples. 

During the synthesis under hydrothermal conditions; There are various factors that can contributed to avoid the adsorption of CO3 or CO2 in our case:

  1. i) First, it´s related to the methodology itself because the used technique: “microwave assisted hydrothermal method” the autoclaves were hermetically close during the reaction.
  2. ii) The Kinetic reactions its very fast under hydrothermal condition, the reaction time was very short in comparison with the other methods, even with conventional hydrothermal technique.
  3. iii) We did used different starting materials than the other researchers, particularly the 2- propanol in this reaction acts as buffer to prevent the formation of different phosphates species because in the current reaction we used sodium tripolyphosphate (Na5P3O10) as phosphate resource; this reagent under aqueous solution easily can be hydrolyzed and simultaneously in presence of Ca2+ solution can form dicalcium phosphate anhydrous (DCPA), which is insoluble. So, the presence of  2-propanol acts as buffer  and  to ensure que  formation of the Silicon hydroxyapatite, we add NH3 to keep pH10. After the reaction the 2-Propal remain in the mother liquor or aqueous supernatant of the reaction and did not react with the HAp. Also, the high solubility of the TMAS in the  aqueous alkaline solution.
  1. Line 153 is TOPAZ 4.2 shouldn't it be TOPAS 4.2?

Thank you for your comment and accept our apologies for the mistake. We correct in the way you have suggested.

Finally, Thank very much for your time and all the suggestion that were consider valuable to improve the manuscript

Reviewer 3 Report

General comments:

I) Several statements were not supported by references.

II). Materials and methods were insufficiently delineated.

III). It was unclear if several independent measurements were determined to assess reproducibility.

IV) It was unclear if Si was inserted into the lattice of of apatites or was adsorbed on the surface of apatites.

V) There was no control on the effects of tetramethylammonium, or of 2 propanol on X-ray powder diffraction, IR, Raman amd morphology on the formation of apatites, especially with respect of OH groups, crystallinity and morphologies.

VI) Overall findings appear preliminary due to lack of controls and reproducibility.

Minor comments:

1) Replace hydroxyapatites by apatites in “The synthesis of partially substituted silicon hydroxyapatite (Si-HAp) nanopowders was systematically investigated via the microwave-assisted hydrothermal process.”

2) Abstract, lines 20-21, p1: Indicate in which medium it was solubilized in “The concentration of the TMAS solutions used varied between 20 0.3 and 1.8 M, corresponding to saturation levels of 1.5–9.0-fold.”

3) Introduction, lines 35-37, p1: Specify HAp in the main text, replace hydroxyapatites by apatites, and rephrase “The preparation of biomaterials with similar characteristics to biological hydroxyapatite, in terms of their chemical and physical properties, involves the uptake of 36 cations and anions in the hexagonal HAp structure.”

4) Introduction, lines 44-46, p1: Delete conventional in “Hitherto, Si-HAp bioceramics were prepared by the conventional method with solid-state reaction at 1000 °C for 6 h, employing β-tricalcium phosphate (β-Ca3(PO4)2), silicon dioxide (SiO2) and calcium carbonate (CaCO3).”

5) Introduction, lines 49-50, p2: Specify sol-gel in “Moreover, chemical solution methods are an alternative for producing both HAp and 49 Si-HAp nanoparticles, such as coprecipitation, neutralization, and sol-gel [6-11].”

6) Introduction, lines 51-52, p2: Add references to support “According to the literature survey, the increase of silicon incorporation in the HAp 51 structure provokes a marked decrease in the crystallite size.”

7) Introduction, lines 52-53, p2: Add references to support “Likewise, silicon affects the particle morphology during the embryo precipitation and particle growth processes.”

8) Introduction, lines 54-55, p2: Add references to support “Recently, Si4+ precursor reagent has been the subject of investigation by a number of 54 research groups worldwide.”

9) Introduction, lines 59-60, p2: Indicate in which medium Si was solubilized, and add references to support “The maximum efficiency of the Si4+ incorporation in the apatite 59 structure was 90 % according to the nominal stoichiometric content of 8.0 mol%.”

10) Introduction, lines 60-62, p2: Specify liquor, and add references to support “Wet chemical quantitative analyses revealed the presence of silicon ions hydrolyzed in the mother liquor.”

11) Introduction, lines 62-63 , p2: Add references to support “In contrast, the challenge of producing synthetic Si-HAp has been carried 62 out by various methods, which also include soft chemistry processes.”

12) Introduction, lines 62-63 , p2: Add references to support “Nowadays, the hydrothermal processing method has demonstrated further chemical reactivity advantages, to yield highly crystalline products with nanometric size.”

13) Introduction, lines 64-66, p2: Add references to support “The reaction kinetics enhancement makes this method efficient for synthesizing materials using relatively low temperatures (100–250 °C).”

14) Introduction, lines 71-73, p2: Specify medium in which it was solubilized, and add references to support “However, the treatments conducted at 200 °C for 8 h limited the incorporation to only 8.0 mol% Si4+, regardless of the nominal stochiometric amount intended (9.0 mol%).”

15) Introduction, lines 73-77, p2: Specify medium in which it was solubilized, and add references to support “Experiments were conducted using the chemical reagents (NH4)3PO4 73 and TEOS as precursor of PO43- and SiO44- However, the uptake of Si4+ was further limited by using (NH4)2HPO4 to 7.65 mol%. The partially substituted Si-HAp particles also incorporate CO32- ions, and the presence of these ions is reported to hinder the incorporation of SiO44- during the crystallization and particle coarsening steps.”

16) Introduction, lines 77-80 p2: Add references to support “Similar results were recently reported by some of the present authors, who attempted the synthesis of Si-HAp under hydrothermal conditions at 150 °C for 10 h employing tetramethyl ammonium silicate ((C4H13NO5Si2), TMAS).”

17) Introduction, lines 80-85, p2: Apatites are difficult to be formed at high pH, delete or rephrase, and add references to support “The low silicon reactivity in the hydrothermal alkaline medium at a pH of 10 caused a limited Si4+ content in the HAp structure of 30 mol% regarding the stoichiometric amount selected (1–20 mol% Si). In this case, the incorporation of CO32- ions was not the cause of the significant Si uptake. The high solubility of the TMAS in the alkaline solution is likely to produce Si complex ions that are highly stable in the hydrothermal medium.”

18) Introduction, lines 86-90, p2: Rephrase “A similar trend was found in the preparation of Zn-substituted HAp, where the isomorphous incorporation of Zn in the Ca site in the HAp structure was affected by the formation of Zn(OH)xn+ species, which are stable in alkaline hydrothermal fluids at the standard pH conditions required to crystallize the HAp [20].”

19) Introduction, lines 91-92, p2: Add references to support “According to the literature survey, the effect of the complex ion formation associated 91 with the dopant ions in HAp has not yet been evaluated in detail.”

20) Introduction, lines 95-97, p2: delete “new” in “Therefore, the present research work is a new approach devoted to investigating the chemical reaction pathway in Si4+ saturated solutions under hydrothermal conditions assisted by microwave heating.”

21) Materials, lines 107-110, p3: It seems that the aqueous solution was not buffered, which could impact the formation of apatites, alternatively indicate the pH of the aqueous (?) solution in “Calcium nitrate tetrahydrate (Ca(NO3)2•4H2O) was used to produce the 1 M solution, while the P5+ precursor selected was sodium tripolyphosphate (Na5P3O10) and the concentration of this solution was 0.2 M”

22) Materials, lines 112-114, p3: Indicate pH of the aqueous solution in “This reagent was used to produce three different solutions with concentrations of 0.3, 0.9 and 1.8 M, which are saturated in Si4+ in comparison with the stoichiometric of 0.2 M.”

23) Materials, lines 112-114, p3: How the stoichiometry was controlled under saturated condition? in “This reagent was used to produce three different solutions with concentrations of 0.3, 0.9 and 1.8 M, which are saturated in Si4+ in comparison with the stoichiometric of 0.2 M.”

24) Materials, lines 114-116, p3 Add reference to support “The organic alcohol 2-propanol (C3H7OH) was used to control the pH of the hydrothermal medium.

25) Microwave-assisted hydrothermal synthesis, lines 118-119, p3: How the stoichiometry was controlled under saturated condition?, specify % in what medium In ” The silicon content selected to produce Ca10(PO4)6-x(SiO4)x(OH)2-x was 6, 10 and 20 118 mol%.

26) Microwave-assisted hydrothermal synthesis, lines 119-121, p3: It was unclear why high volume of 2-propanol was used (15 mL), specify pH of the solution in “The standard processing involved the preparation of a mother solution constituted 119 by 17.5 ml of the 1 M Ca2+ solution and 15 ml of 2-propanol, and this solution was 120 magnetically stirred for 5 min.”

27) Microwave-assisted hydrothermal synthesis, lines 119-121, p3: Indicate final concentrations of Ca, P and 2-propanol, as well as pH in “In parallel, a solution mixture (17.5 ml) containing P5+ and 121 Si4+ ions was prepared according to the molar mixing Ca/(P+Si) ratio of 1.67.”

28) Microwave-assisted hydrothermal synthesis, lines 122-124, p3: Rephrase since there were three compounds Ca, P, and Si for ratio of two compounds in “Therefore, the mixtures volumes calculated by the molar ratio Ca:P:Si were 17.5:0, 16.45:1.05, 15.75:1.75, 14.0:3.5,”

29) Microwave-assisted hydrothermal synthesis, lines 122-124, p3: Rephrase” The colloidal suspension pH was adjusted to a value of 10.00 129 ± 0.1 by adding dropwise a 1.0 M NH4OH solution [1,14,20].”

30) Microwave-assisted hydrothermal synthesis, lines 129-130, p3: There is an increase in the formation of the monoclinic phase and a decrease of the hexagonal phase when the pH value diminishes from 9.6 to 7. as reported by S. López-Ortiz,D. Mendoza-Anaya,D. Sánchez-Campos,M. E. Fernandez-García,E. Salinas-Rodríguez,M. I. Reyes-Valderrama, and V. Rodríguez-Lugo. The pH Effect on the Growth of Hexagonal and Monoclinic Hydroxyapatite Synthesized by the Hydrothermal Method. Journal of Nanomaterials, vol. 2020, Article ID 5912592, 10 pages, 2020. https://doi.org/10.1155/2020/5912592. It was unclear why colloidal suspension was adjusted at pH= 10, especially the aim was to prepare biomaterials with similar characteristics to biological hydroxyapatite in “The colloidal suspension pH was adjusted to a value of 10.00 129 ± 0.1 by adding dropwise a 1.0 M NH4OH solution [1,14,20].”

31) Microwave-assisted hydrothermal synthesis, lines 130-132, p3: It seems that there was 17.5 mL of calcium solution and 15 mL of 2-propanol and it was unclear what was the final composition, rephrase “The suspension (50 ml) was then transferred to a double-walled Teflon high pressure vessel and placed in the rotatory device of the microwave oven (MARS-5X, CEM Corp., USA).”

32) Microwave-assisted hydrothermal synthesis, lines 136-140, p3: The equation was unbalanced with respect of Na+, NO3-, CH3CH2OH, C4H13NO5, SIO44-, CH4, etc… The number of X and Y were unbalanced. The whole equation was not neutral in “In Equation (1), the ultimate content of (Si-O-Si)3O- in the reaction products is equivalent to the subtraction of the Si4+ incorporated in the HAp and the nominal content supplied. Furthermore, OH- deficiency in Ca10(PO43-)6-x(SiO44-)x(OH)2-x results from the charge balance required to compensate the total negative valence of SiO44- groups incorporated in the HAp structure.”

33) Characterization, line 148, p4: Delete “The residual crystalline phases were determined by X-ray powder diffraction (XRD) 1analyses.”

34) Characterization, line 151-154, p4: Add references to support “Furthermore, Rietveld refinement analyses of selected samples were carried out to determine the crystallite size and lattice parameters using the TQPAZ 4.2 software. ”

35) Characterization, lines 155-156,p4: Delete “Fourier transform infra-red (FT-IR) analyses were carried out to determine aspects of 155 the vibrational OH- modes and functional PO43- group.”

36) Results and Discussion, line 182, p4: Results and discussion shall be separated.

37) Effect of Si4+ saturation on the hydrothermal synthesis of Si-Hap, lines 198-203, p5: It was unclear how reproducible was the XRD patterns in “The sample prepared with the 0.3 M TMAS solution with the volume to supply 0.33 mol% Si exhibited a slight shifting of the peak to a lower diffraction angle (Figure 1a). ). In contrast, at 4.5 and 9-fold saturation levels, a progressive displacement of the XRD pattern proceeded at small 2θ angles, and also a remarkable peak broadening occurred on the diffraction patterns, as shown in Figures 1(b-c).”

38) Crystalline structural and chemical compositional analyses of Si-HAp powders, lines 220-228, p5: It was unclear what was the sample variations to support “The Rietveld refinement approach provided the lowest values of the goodness-of-fit factor (GOF/Chi2), averaging 1.23 ± 0.6 and low Rwp 7.32± 1.0 values (Table 1), which confirm the excellent accuracy of the refinement approach investigated in our case. Figures 2(a-b) show the calculated and experimental diffraction profiles together with the residual fitting curve obtained. The lower trace is the difference between the observed and calculated patterns depicted by the residual straight line, and the vertical lines correspond to the calculated Bragg peaks positions. It is worth mentioning the good agreement between both.”

39) Crystalline structural and chemical compositional analyses of Si-HAp powders, Table 1, Lines 252-253, p6: Indicate number of independent measurements, standard errors TMA concentrations, and specify or detete MM62, MM66, MM63, MM50, MM51, MM53, MM64, MM68, MM67 in “Table 1. Chemical and physical features of Si-HAp powders prepared under hydrothermal assisted microwave treatments at 150 °C, 1 h using different concentrations of TMAS solutions and tripolyphosphate as P precursor.”

40) Crystalline structural and chemical compositional analyses of Si-HAp powders,   Lines 259-261, p7: What was the reproducibility in the intensity of the OH group since it was unclear if this can also be associated to water, and specify vibrational mode located at 631 cm-1 to support “Other crystalline features of the Si-HAp powders were examined by FT-IR analyses. 259 Figures 3(a-c) show the results of the analysis for HAp and Si-HAp powders prepared 260 under the microwave-assisted hydrothermal technique, which reveals hydroxyl (OH-) 261 groups stretching (3571 cm-1) and vibration (631 cm-1) bands.”

41) Crystalline structural and chemical compositional analyses of Si-HAp powders, Lines 277-281, p7: Delete “Additionally, the intensity of the OH- group band on the 0.16 and 0.90 mol% Si samples was similar, and this was irrespective of the saturation contents of Si+4 provided by the TMAS solutions of 0.3 and 0.9 M. On the contrary, the OH- stretching band at 3571 cm-1 and 960 cm-1 is markedly reduced in Si-HAp powders prepared with the highest concentration 1.8 M of TMAS (Figure 3c).

42) Crystalline structural and chemical compositional analyses of Si-HAp powders, Lines 281-284 and lines 285-287, p7: I am not convinced that the variations in XRD and IR spectra indicated solely insertion of Si into the apatite lattice. There is the possibility that suspension due to increased amount of Si induced less crystalline apatites, which could produce variable XRD and IR. The possibility that Si interacted on the surface of mineral complexes can’t be ruled out. Delete ”These structural variations together with the XRD analyses confirm that a bulky Si incorporation took place in the apatite structure rather than partially at the particle surface as reported elsewhere [26].” and “These structural results are consistent with the substitution mechanism proposed, where PO4-3 ions are replaced by SiO44− ions, causing a stoichiometric release of OH- ions, which maintains the total charge balance in the HAp structure (Equation 1).”

43) Crystalline structural and chemical compositional analyses of Si-HAp powders, Lines 287-292, p7: Specify V in “Under hydrothermal conditions the saturation of Si4+ probably inducing a reduction in the amount of hydroxyl groups to compensate an extra negative electric charge produced by the incorporation of the silicate groups, and the formation of OH- vacancies might take place to maintain the charge balance neutrality, as is described by the following equation: PO43— + OH- → SiO44− + V(OH)-.”

44) Crystalline structural and chemical compositional analyses of Si-HAp powders, Figure 3, Lines 294-300, p7: Replace CH3 by OH in each spectra located at around 1650 cm-1 “Figure 3. FT-IR analysis of the Si-HAp powders obtained by assisted hydrothermal process at 150 °C for 1 h using different 294 concentrations of [(C4H13NO5Si2) (TMAS)]: (a) 0.3 M; (b) 0.9 M and (c) 1.8 M.”

45) Crystalline structural and chemical compositional analyses of Si-HAp powders, Lines 303-304, p7: XRD does not indicate any information on OH groups, while IR is sensitive to water band, rephrase “Likewise, the OH- peak decreased with broadening of the band, as was mentioned elsewhere [20]; these results also support the FT-IR results and XRD results.”

46) Crystalline structural and chemical compositional analyses of Si-HAp powders, Lines 311-315, p9 : Alternatively IR and Raman samples were not checked for reproducibility, which may explain distinct findings from single IR and Raman measurements, rephrase “However, Si-HAp samples synthesized with a large content of Si4+ (12.16 mol%) did not show the presence of the characteristic Si signal, and only small changes were detected in the vibration OH- signal at 3570 cm-1, which were slightly decreased and broadened under the high Si4+ saturation of TMAS (1.8 M) during the hydrothermal reaction.”

47) Crystalline structural and chemical compositional analyses of Si-HAp powders, Lines 311-315 and 343-353, p9: Itwas unclear if the findings were reproducible in “However, the presence of silicon was not determined in powders prepared with 0.41 336 mol% Si, as shown in Figure 5d. In contrast, a symmetric peak corresponding to the core 337 level Si 2p at BE of 103.3 eV was revealed in the Si-HAp samples obtained with the molar volume corresponding to 5.0 mol% Si using the TMAS solution of 0.9 M.” and in “ By way of contrast, experiments conducted employing the highest Si4+ saturated TMAS solution (1.8 M) exhibit a gradual uptake of Si, which was revealed by the XPS spectra in Figure 6 corresponding to HAp and Si-HAp powders prepared under the microwave-assisted hydrothermal processing. All the samples are constituted by the chemical elements that form the HAp structure. Figure 6a shows the typical doublet peak associated with the core level Ca 2p3/2 XPS BE at 347.1 eV and Ca 2p1/3 at 350.07 eV. The doublet peaks exhibited a slight increase as the silicon incorporation content increased in the synthesized samples obtained with 1.61 mol% of Si in the Si-HAp powders. Likewise, the symmetric signals of P 2p exhibited a slight increase in intensity with the increase of Si4+ content, achieving a binding energy of 132.92 eV. Furthermore, the binding energy signals for O 2p peaks are nearly symmetric (Figure 6c).”

48) Morphological aspects of the partially substituted Si-HAp particles prepared hydrothermally, lines 400-401, p11: Add statistically analysis of diameter sizes of samples to support “These results are supported by variation in the crystallite size, as calculated in the Rietveld refinement results (Table 1).”

49) Conclusion, lines 465-466, p13: It was unclear how % of Si was determined in “The maximum amount of Si4+ incorporated in the HAp structure was 12.16 mol%, using an excess of 1.8 M of TMAS.”

50) Conclusion, lines 469-474 , p13: It was unclear if the increased concentration of tetramethylammonium instead of Si moiety induced agglomerates in “The Si4+ excess in the reaction media caused the rod-like crystal self-assembly to produce irregular oval-shaped Si-HAp agglomerates, which were prepared under fast kinetic reaction conditions assisted by the microwave heating and exhibit sizes between 233.5 and 315.1 nm. These agglomerates exhibited a marked size coarsening, which was triggered by the Si4+ saturation level supplied in the hydrothermal media.”

51) Conclusion, lines 478-480, and lines 481-484, p13: Delete “Furthermore, a remarkable decrease on the OH- ions content was confirmed by FTIR and XPS analyses, the gradual OH- lost was caused to compensate the partial incorporation of SiO44- at tetrahedral PO43- sites in the HAp structure.” And “The present hydrothermal microwave assisted method has delivered high processing efficiency to crystallize Si-HAp particles with a control on the Si4+ content. This method has potential for processing Si-HAp bioceramic implants in medicine.”

Author Response

III. Response to Reviewer 3

Thank you for the time and comments that are relevant to improve the manuscript technical quality.

1) Replace hydroxyapatites by apatites in “The synthesis of partially substituted silicon hydroxyapatite (Si-HAp) nanopowders was systematically investigated via the microwave-assisted hydrothermal process.”

Thank you very much for the time spent to review our manuscript. We found all the comments useful to improve the fundamentals in the document, and we entirely address all the comments made to our preliminary manuscript version.

In the present research it’s been preparing synthetic Substitute Silicon hydroxyapatite and into the hexagonal structure still content OH- ion inside the structure Ca10(PO43-)6-x(SiO44-)x(OH)2-x(s).

In many papers: Hydroxyapatite, it is called hydroxyapatite (HAp or HA), for naturally occurring mineral form of calcium apatite with the formula Ca5(PO4)3(OH), but is usually written Ca10(PO4)6(OH)2 to denote the crystal unit cell .

Normally, it’s found that the term of Apatite is more common for natural and biological apatite rather than synthetic hydroxyapatite, that is why we did used the term of Ca10(PO4)6(OH)2 “Hydroxyapatite”.

2) Abstract, lines 20-21, p1: Indicate in which medium it was solubilized in “The concentration of the TMAS solutions used varied between 20 0.3 and 1.8 M, corresponding to saturation levels of 1.5–9.0-fold.”

Thank you for your comment in section 2.1 is mentioned that all the reagent are dissolved in distillated water, and I did added the word in the line 20-21, “aqueous” as you suggest.

3) Introduction, lines 35-37, p1: Specify HAp in the main text, replace hydroxyapatites by apatites, and rephrase “The preparation of biomaterials with similar characteristics to biological hydroxyapatite, in terms of their chemical and physical properties, involves the uptake of 36 cations and anions in the hexagonal HAp structure.”

Thank you very much for the suggestion, It was modify in manuscript as follows:

The preparation of biomaterials with similar chemical and physical properties to biological hydroxyapatite (HAp), in terms of their chemical and physical properties, involves the uptake of cations and anions in the hexagonal HAp structure. The incorporation of Si4+ ions into the PO43- unit network of the HAp stimulates both bone formation and resorption processes, which are relevant to both tissue restauration and bone growth [1].

4) Introduction, lines 44-46, p1: Delete conventional in “Hitherto, Si-HAp bioceramics were prepared by the conventional method with solid-state reaction at 1000 °C for 6 h, employing β-tricalcium phosphate (β-Ca3(PO4)2), silicon dioxide (SiO2) and calcium carbonate (CaCO3).”

Thank you again for your kind suggestion and comments: It was modify the paragraph as follows:

Hitherto, Si-HAp bioceramics were prepared with a conventional solid-state reaction at 1000 °C for 6 h, employing β-tricalcium phosphate (β-Ca3(PO4)2), silicon dioxide (SiO2) and calcium carbonate (CaCO3). These partially silicon substituted powders exhibit a good biological dissolution capability in comparison with pure HAp [5,12].

5) Introduction, lines 49-50, p2: Specify sol-gel in “Moreover, chemical solution methods are an alternative for producing both HAp and 49 Si-HAp nanoparticles, such as coprecipitation, neutralization, and sol-gel [6-11].”

Thanks for your comment, It was modify in manuscript as follows:

Moreover, chemical solution methods, such as coprecipitation, neutralization, and sol-gel are an alternative synthetic procedures for producing both HAp and Si-HAp nanoparticles. The increase of silicon incorporation in the HAp structure provokes a marked decrease in the crystallite size [6-11].

6) Introduction, lines 51-52, p2: Add references to support “According to the literature survey, the increase of silicon incorporation in the HAp 51 structure provokes a marked decrease in the crystallite size.”

Thank you for the suggestion, and comment, It was modify in manuscript as follows and add the reference:

Likewise, silicon affects the particle morphology during the embryo precipitation and particle growth processes. Recently, extensive attentions have been paid to the appropriated process with Si4+ precursor reagent to overcome the difficulties in handing, associated with its reactivity under wet chemical processing [4-15].

7) Introduction, lines 52-53, p2: Add references to support “Likewise, silicon affects the particle morphology during the embryo precipitation and particle growth processes.”

Thank you, for the suggestion, however already the paragraph was modify as follow:

Likewise, silicon affects the particle morphology during the embryo precipitation and particle growth processes. Recently, extensive attentions have been paid to the appropriated process with Si4+ precursor reagent to overcome the difficulties in handing, associated with its reactivity under wet chemical processing [4-15].

8) Introduction, lines 54-55, p2: Add references to support “Recently, Si4+ precursor reagent has been the subject of investigation by a number of 54 research groups worldwide.”

Thank you for your comment and we correct in the way you have suggested.

Likewise, silicon affects the particle morphology during the embryo precipitation and particle growth processes. Recently, extensive attentions have been paid to the appropriated process with Si4+ precursor reagent to overcome the difficulties in handing, associated with its reactivity under wet chemical processing [4-15].

9) Introduction, lines 59-60, p2: Indicate in which medium Si was solubilized, and add references to support “The maximum efficiency of the Si4+ incorporation in the apatite 59 structure was 90 % according to the nominal stoichiometric content of 8.0 mol%.”

Thank you for your comment and It was correct in the way you have suggested :

Hitherto, tetraethyl orthosilicate (Si(OCH2CH3)4, TEOS) in polyethylene glycol/water and silicon tetra-acetate in water (Si(CH3CO2)4) have been better reagents for incorporating Si4+ in the hexagonal structure [13]. The maximum efficiency of the Si4+ incorporation in the apatite structure was 90 % according to the nominal stoichiometric content of 8.0 mol%. Wet chemical quantitative analyses revealed the presence of silicon ions hydrolyzed in the remaining mother liquor after precipitation of SiHAp. In contrast, the challenge of producing synthetic Si-HAp has been carried out by various methods, including soft chemistry processes [5-13].

10) Introduction, lines 60-62, p2: Specify liquor, and add references to support “Wet chemical quantitative analyses revealed the presence of silicon ions hydrolyzed in the mother liquor.”

Thank you for your comment and It was correct in the way you have suggested. Here, the term mother liquid is refer to express the remanent liquid after the reaction

Wet chemical quantitative analyses revealed the presence of silicon ions hydrolyzed in the remaining mother liquor after precipitation of SiHAp. In contrast, the challenge of producing synthetic Si-HAp has been carried out by various methods, including soft chemistry processes [5-13].

11) Introduction, lines 62-63 , p2: Add references to support “In contrast, the challenge of producing synthetic Si-HAp has been carried 62 out by various methods, which also include soft chemistry processes.”

Thank you for your comment and It was correct in the way you have suggested:

“In contrast, the challenge of producing synthetic Si-HAp has been carried out by various methods, including soft chemistry processes [5-13].”

12) Introduction, lines 62-63 , p2: Add references to support “Nowadays, the hydrothermal processing method has demonstrated further chemical reactivity advantages, to yield highly crystalline products with nanometric size.”

Thank you very much again for your suggestion, It was correct in the way you have suggested:

“The hydrothermal process has brought further advantages in terms of chemical reactivity: higher yield for crystalline products with nanometric size and the reaction kinetics reaction kinetics enhancement even at relatively low temperatures (100–250 °C), [14, 17,21].”

13) Introduction, lines 64-66, p2: Add references to support “The reaction kinetics enhancement makes this method efficient for synthesizing materials using relatively low temperatures (100–250 °C).”

Thank you for your comment and It was correct in the way you have suggested, and its included also reference  was added.

The hydrothermal process has brought further advantages in terms of chemical reactivity: higher yield for crystalline products with nanometric size and the reaction kinetics reaction kinetics enhancement even at relatively low temperatures (100–250 °C), [14, 17,21].

14) Introduction, lines 71-73, p2: Specify medium in which it was solubilized, and add references to support “However, the treatments conducted at 200 °C for 8 h limited the incorporation to only 8.0 mol% Si4+, regardless of the nominal stochiometric amount intended (9.0 mol%).”

Thank you very much for your kind observation and comment: we added the reference:

However, on the synthesis conducted at 200 °C for 8 h limited the incorporation to only 8.0 mol% Si4+, regardless of the nominal stochiometric amount intended (9.0 mol%) [13].

15) Introduction, lines 73-77, p2: Specify medium in which it was solubilized, and add references to support “Experiments were conducted using the chemical reagents (NH4)3PO4 73 and TEOS as precursor of PO43- and SiO44- However, the uptake of Si4+ was further limited by using (NH4)2HPO4 to 7.65 mol%. The partially substituted Si-HAp particles also incorporate CO32- ions, and the presence of these ions is reported to hinder the incorporation of SiO44- during the crystallization and particle coarsening steps.”

Thank you for your comment and It was correct in the way you have suggested

In other experiments , the uptake of Si4+ was further limited by using (NH4)2HPO4 to 7.65 mol%. The partially substituted Si-HAp particles also incorporate CO32- ions, and the presence of these ions is reported to hinder the incorporation of SiO44- during the crystallization and particle coarsening steps[13].

16) Introduction, lines 77-80 p2: Add references to support “Similar results were recently reported by some of the present authors, who attempted the synthesis of Si-HAp under hydrothermal conditions at 150 °C for 10 h employing tetramethyl ammonium silicate ((C4H13NO5Si2), TMAS).”

Thank you for your comment and It was correct in the way you have suggested In this case we added the reference, and the paragraph now is as following:

Similar experiments were recently conducted to attempted the synthesis of Si-HAp under hydrothermal conditions at 150 °C for 10 h by employing tetramethyl ammonium silicate ((C4H13NO5Si2), TMAS). The low silicon reactivity in the hydrothermal alkaline medium at a pH of 10 caused a limited Si4+ content in the HAp structure of 30 mol% regarding the stoichiometric amount selected (1–20 mol% Si). In this case, the incorporation of CO32- ions was not the cause of the significant Si uptake. The high solubility of the TMAS in the alkaline solution is likely to produce Si complex ions that are highly stable in the hydrothermal medium, giving rise to the decrease in the Si concentration in the embryo and growth steps [20].

17) Introduction, lines 80-85, p2: Apatites are difficult to be formed at high pH, delete or rephrase, and add references to support “The low silicon reactivity in the hydrothermal alkaline medium at a pH of 10 caused a limited Si4+ content in the HAp structure of 30 mol% regarding the stoichiometric amount selected (1–20 mol% Si). In this case, the incorporation of CO32- ions was not the cause of the significant Si uptake. The high solubility of the TMAS in the alkaline solution is likely to produce Si complex ions that are highly stable in the hydrothermal medium.”

Thank you for your comment and It was correct in the way you have suggested:

The low silicon reactivity in the hydrothermal alkaline medium at a pH of 10 caused a limited Si4+ content in the HAp structure of 30 mol% regarding the stoichiometric amount selected (1–20 mol% Si). In this case, the incorporation of CO32- ions was not the cause of the significant Si uptake. The high solubility of the TMAS in the alkaline solution is likely to produce Si complex ions that are highly stable in the hydrothermal medium, giving rise to the decrease in the Si concentration in the embryo and growth steps [20].

18) Introduction, lines 86-90, p2: Rephrase “A similar trend was found in the preparation of Zn-substituted HAp, where the isomorphous incorporation of Zn in the Ca site in the HAp structure was affected by the formation of Zn(OH)xn+ species, which are stable in alkaline hydrothermal fluids at the standard pH conditions required to crystallize the HAp [20].”

Thank you for your comment and It was correct in the way you have suggested

A similar trend was found in the preparation of Zn-substituted HAp, where the isomorphous incorporation of Zn at the Ca site in the HAp structure was affected by the formation of Zn(OH)xn+ species, which were  also stables in alkaline hydrothermal fluids at the standard pH conditions required to crystallize the HAp [21].

19) Introduction, lines 91-92, p2: Add references to support “According to the literature survey, the effect of the complex ion formation associated 91 with the dopant ions in HAp has not yet been evaluated in detail.”

Thank you for your comment and we made the changing. It was correct in the way you have suggested and the reference was added.

Although the detailed effect of the complex ion formation associated with the dopant ions in HAp has not been evaluated yet, it is important from the chemical processing point of view, to enhance the control of the stoichiometry of Ca10(PO4)6-x(SiO4)x(OH)2-x solid solutions and the particle growth at nanometer order [20].

20) Introduction, lines 95-97, p2: delete “new” in “Therefore, the present research work is a new approach devoted to investigating the chemical reaction pathway in Si4+ saturated solutions under hydrothermal conditions assisted by microwave heating.”

Thank you for your comment and It was correct in the way you have suggested

In the present research work, different approaches for synthesis of Ca10(PO4)6-x(SiO4)x(OH)2-x particle were investigated devoted to investigating the chemical reaction pathway in Si4+ saturated solutions under hydrothermal conditions assisted by microwave heating.

21) Materials, lines 107-110, p3: It seems that the aqueous solution was not buffered, which could impact the formation of apatites, alternatively indicate the pH of the aqueous (?) solution in “Calcium nitrate tetrahydrate (Ca(NO3)2•4H2O) was used to produce the 1 M solution, while the P5+ precursor selected was sodium tripolyphosphate (Na5P3O10) and the concentration of this solution was 0.2 M”

Thank you for your comments, now in the Material Sections was describe on page 110-118 is : However,  the intention of the 2-propanol addition was because during the hydrothermal reaction  the protons generate with the hydrolysis of calcium tripolyphosphate to orthophosphate ions . during the  formation of HAp, an alkali agent ( NH3) was added to the initial gel to neutralize the proton and to prevent the formation of the dicalcium phosphate anhydrous (DCPA). This phosphate has low solubility among calcium phosphate compound  compounds in the range of pH<4. In consequence, in our experiments the pH was adjusted 9.8-10.0 with NH3 as a alkali reagent. Reference [28]  .

Preparation of the reagents to synthesis of the stoichiometric pure hydroxyapatite (HAp) and silicon substituted hydroxyapatite (Si-HAp) powders was carried out as follows; All the chemicals of reagent grade (Sigma Aldrich, 99.99% purity) were used without further purification. The 1M Ca2+ and 0.2M P5+ stock solutions were prepared by dissolving calcium nitrate tetrahydrate (Ca(NO3)2•4H2O) and sodium tripolyphosphate (Na5P3O10) in distilled water, respectively. The Si4+ stock solutions of three different concentrations of 0.3, 0.9 and 1.8 M were prepared by dissolving was tetramethylammonium silicate solution [(C4H13NO5Si2) (TMAS)]. Also, all aqueous TMAS solutions were adjusted to pH 10 with 7M NH3 solution before making up to final volume of the TMAS stocks. The 7M of NH3 stock solution was prepared by mixing 82.6 ml of conc. NH3solution with 17.35 ml of water. 2-Propanol was added as buffer to prevent the formation of another phosphorous  species during the reaction.

22) Materials, lines 112-114, p3: Indicate pH of the aqueous solution in “This reagent was used to produce three different solutions with concentrations of 0.3, 0.9 and 1.8 M, which are saturated in Si4+ in comparison with the stoichiometric of 0.2 M.”

Thank you very much for the comment; It was correct in the way you have suggested

Certainly the concentration of the solution of TMAS  used as Si 4+ precursor was prepared with tree different concentration, 03, 0.9, and 1.9 M  which it means: different level of saturation of Si4+: 1.5, 3 and 9 fold. For all aqueous TMAS solutions were adjusted to pH=10 with 7M NH3 solution before making up to final volume of the TMAS stocks.

23) Materials, lines 112-114, p3: How the stoichiometry was controlled under saturated condition? in “This reagent was used to produce three different solutions with concentrations of 0.3, 0.9 and 1.8 M, which are saturated in Si4+ in comparison with the stoichiometric of 0.2 M.”

Thank you for your comment and It was correct in the way you have suggested

In line 110- 111;  The stoichiometric is controlling because initial Mix of the all starting reagents was adjusted the pH to 10 and immediately the chamber of the autoclaves were  hermetically closed and after that was conducted the hydrothermal treatment  .

24) Materials, lines 114-116, p3 Add reference to support “The organic alcohol 2-propanol (C3H7OH) was used to control the pH of the hydrothermal medium.

Thank you for your comment and It was correct in the way you have suggested: The reference was added It was suggested

The pH was adjusted to 10 with 7M of NH3 solution.  2-propanol was used as buffer to control the pH during the reaction:

As it was explain above; The 2-propanol addition was because during the hydrothermal reaction  the protons generate with the hydrolysis of calcium tripolyphosphate to orthophosphate ions . during the  formation of HAp, an alkali agent (NH3) was added to the initial gel to neutralize the proton and to prevent the formation of the dicalcium phosphate anhydrous (DCPA). This phosphate has low solubility among calcium phosphate compound  compounds in the range of pH<4. In consequence, in our experiments the pH was adjusted 9.8-10.0 with NHas a alkali reagent. Reference  [28]  .

25) Microwave-assisted hydrothermal synthesis, lines 118-119, p3: How the stoichiometry was controlled under saturated condition?, specify % in what medium In ” The silicon content selected to produce Ca10(PO4)6-x(SiO4)x(OH)2-x was 6, 10 and 20 118 mol%.

Thank you very much for your kind question: The answer is as follows:

Because the reaction system is in aqueous solution;  initially it was fix the initial stoichiometric of the reaction by calculate the appropriated  amount of each starting reagent and adjusted the initial pH in 10. For all the experiments the pH was checked at the initial  and the final steps to ensure the appropriated control of the alkali in the aqueous medium during the reaction.

26) Microwave-assisted hydrothermal synthesis, lines 119-121, p3: It was unclear why high volume of 2-propanol was used (15 mL), specify pH of the solution in “The standard processing involved the preparation of a mother solution constituted 119 by 17.5 ml of the 1 M Ca2+ solution and 15 ml of 2-propanol, and this solution was 120 magnetically stirred for 5 min.”

Thank you very much for your kind question and comment:

As I did mention on questions 21 and 24  now lines 119-120, the 2- propanol in this reaction acts as buffer to prevent the formation of different phosphates species because in the current reaction we used  sodium tripolyphosphate (Na5P3O10) as  resource of phosphate , this reagent under aqueous solution easily can be hydrolyzed and simultaneously in presence of Ca2+ solution can form dicalcium phosphate anhydrous (DCPA), which is insoluble. So, the presence of  2-propanol acts as buffer  and  to ensure que  formation of the Silicon hydroxyapatite, we add NH3 to keep pH10. After the reaction the 2-Propal remain in the mother liquor or aqueous supernatant of the reaction.

27) Microwave-assisted hydrothermal synthesis, lines 119-121, p3: Indicate final concentrations of Ca, P and 2-propanol, as well as pH in “In parallel, a solution mixture (17.5 ml) containing P5+ and 121 Si4+ ions was prepared according to the molar mixing Ca/(P+Si) ratio of 1.67.”

Thank you very much you're your kid question:

The answer is:  Yes, you can find the final concentration of the Ca, P, Si on Table 1.   Normally in this type of hydrothermal reaction the concentration of 2-propanol it is no determinate directly.  We only made sure that after the reaction the final volume corresponding to the mother liquid or aqueous supernatant was 50 ml, same volume as initially.

28) Microwave-assisted hydrothermal synthesis, lines 122-124, p3: Rephrase since there were three compounds Ca, P, and Si for ratio of two compounds in “Therefore, the mixtures volumes calculated by the molar ratio Ca:P:Si were 17.5:0, 16.45:1.05, 15.75:1.75, 14.0:3.5,”

Thank you for your comment and It was correct in the way you have suggested:

A mother solution constituted by 17.5 ml of the 1 M Ca2+ solution and 15 ml of 2-propanol was magnetically stirred for 5 min. The added 2-propanol is used as pH buffer to prevent the hydrolysis of calcium tripolyphosphate gel to orthophosphate ions . In parallel, a solution mixture (17.5 ml) containing P5+and Si4+ ions was prepared according to the molar mixing Ca/(P+Si) ratio of 1.67. Therefore, the mixtures volumes calculated by the molar ratio Ca:P:Si were 17.5:0, 16.45:1.05, 15.75:1.75, 14.0:3.5, where the molar volumes correspond to the pure HAp and the selected silicon compositions of 6, 10 and 20 mol%, respectively.

29) Microwave-assisted hydrothermal synthesis, lines 122-124, p3: Rephrase” The colloidal suspension pH was adjusted to a value of 10.00 129 ± 0.1 by adding dropwise a 1.0 M NH4OH solution [1,14,20].”

Thank you for your comment and It was correct in the way you have suggested:

Them, pH of the colloidal suspension was adjusted to a value of 10.00 ± 0.1 by adding a 7.0 M NH3 aqueous solution dropwisely [1,14,20]

30) Microwave-assisted hydrothermal synthesis, lines 129-130, p3: There is an increase in the formation of the monoclinic phase and a decrease of the hexagonal phase when the pH value diminishes from 9.6 to 7. as reported by S. López-Ortiz,D. Mendoza-Anaya,D. Sánchez-Campos,M. E. Fernandez-García,E. Salinas-Rodríguez,M. I. Reyes-Valderrama, and V. Rodríguez-Lugo. The pH Effect on the Growth of Hexagonal and Monoclinic Hydroxyapatite Synthesized by the Hydrothermal Method. Journal of Nanomaterials, vol. 2020, Article ID 5912592, 10 pages, 2020. https://doi.org/10.1155/2020/5912592. It was unclear why colloidal suspension was adjusted at pH= 10, especially the aim was to prepare biomaterials with similar characteristics to biological hydroxyapatite in “The colloidal suspension pH was adjusted to a value of 10.00 129 ± 0.1 by adding dropwise a 1.0 M NH4OH solution [1,14,20].”

Thank you very much for the kind suggestion: It was correct in the way you have suggested:

On the reference you kindly provided, they made experiments using (NH4)2HPO4 and also it was changing gradually the pH with different types reagents to adjusted the pH to study the effect of the pH on the structure of HAp. 

In contrast, the reaction system of the present research was different than that of you kindly refer. We keep constant the pH, and only change the Silicon concentration.

Actually, we did adjusted the pH to 10±01 in order to be sure the formation of the pure hydroxyapatite (HAp) and silicon hydroxyapatite ( Si-HAp)with hexagonal structure. It was proved on the XRD results as well on the Rietveld refinement. In addition, if the reaction take place without adjusted the pH to 10 under hydrothermal conditions, other subproducts of Ca or Phosphate can be produced. In this case it was used as a starting materials particularly we did use different phosphate source and different silicon source and all of them are dissolved in water.

In addition, In our case we did not used a “big autoclaves” to do our experiments. So I can no compare the system and the structure results of the current  research with this reference.

31) Microwave-assisted hydrothermal synthesis, lines 130-132, p3: It seems that there was 17.5 mL of calcium solution and 15 mL of 2-propanol and it was unclear what was the final composition, rephrase “The suspension (50 ml) was then transferred to a double-walled Teflon high pressure vessel and placed in the rotatory device of the microwave oven (MARS-5X, CEM Corp., USA).”

Thank you for your kind observation and comments : It was correct in the way you have suggested to clarify

“A mother solution constituted by 17.5 ml of the 1 M Ca2+ solution and 15 ml of 2-propanol was magnetically stirred for 5 min. The added 2-propanol is used as pH buffer to prevent the hydrolysis of calcium tripolyphosphate gel to orthophosphate ions . In parallel, a solution mixture (17.5 ml) containing P5+ and Si4+ions was prepared according to the molar mixing Ca/(P+Si) ratio of 1.67. Therefore, the mixtures volumes calculated by the molar ratio Ca:P:Si were 17.5:0, 16.45:1.05, 15.75:1.75, 14.0:3.5, where the molar volumes correspond to the pure HAp and the selected silicon compositions of 6, 10 and 20 mol%, respectively. To investigate the effect of the Si saturation in the mother liquor, the molar volumes calculated were provided with the silicon solutions of 0.3, 0.9 and 1.8 M silicon solutions were selected to investigate the effect of the Si saturation in the mother liquor, respectively. On mixing of both solutions instantaneously, a white milky colloid formed and the colloidal suspension was stirred constantly for 15 min.”

Well, as it was mention :During the hydrothermal experiments: 17.5 ml of  calcium solution plus 15 ml of 2-propanol, in parallel a mix of different volume of P5+ solutions plus different volume of TMAS solutions was stir it, once the both solutions was properly mixed separately, the mixing of  two solution formed a colloidal suspension of the Si-HAP which was constantly stirred.

The final composition of the samples: On Table 1 can found the Chemical Composition of each sample.

“The suspension (50 ml) was then transferred to a double-walled Teflon high pressure vessel and placed in the rotatory device of the microwave oven (MARS-5X, CEM Corp., USA), and was heated at 150 °C for 1 h.”

32) Microwave-assisted hydrothermal synthesis, lines 136-140, p3: The equation was unbalanced with respect of Na+, NO3-, CH3CH2OH, C4H13NO5, SIO44-, CH4, etc… The number of X and Y were unbalanced. The whole equation was not neutral in “In Equation (1), the ultimate content of (Si-O-Si)3O- in the reaction products is equivalent to the subtraction of the Si4+ incorporated in the HAp and the nominal content supplied. Furthermore, OH- deficiency in Ca10(PO43-)6-x(SiO44-)x(OH)2-x results from the charge balance required to compensate the total negative valence of SiO44- groups incorporated in the HAp structure.”

Thank very much for your comment: I do want do apologize for the mistake, now the equations is as following:

10 Ca(NO3)2(aq) + 2(Na5P3O10)(aq)+ y(C4H13NO5Si2)(aq) + xNH4OH(aq)+ x(CH3)2CHOH

Ca10(PO43-)6-x(SiO44-)y(OH)2-x(s) + 10Na+ + xOH(aq) + xNH4d+NO3d (aq) + xH2O +yCH4d+ + (y-x)[(Si-O-Si)3O-)](aq)                                                                                                                      (1)

33) Characterization, line 148, p4: Delete “The residual crystalline phases were determined by X-ray powder diffraction (XRD) 1analyses.”

Thank you for your comment and for suggestions however, I did not understand your suggestion clearly. It was correct in the section: as following: The crystalline phases of the obtained powders were determined by X-ray powder diffraction (XRD) analyses.

34) Characterization, line 151-154, p4: Add references to support “Furthermore, Rietveld refinement analyses of selected samples were carried out to determine the crystallite size and lattice parameters using the TQPAZ 4.2 software. ”

Thank you very much indeed for your comment It was correct in the way you have suggested: the Reference was added [25].

35) Characterization, lines 155-156,p4: Delete “Fourier transform infra-red (FT-IR) analyses were carried out to determine aspects of 155 the vibrational OH- modes and functional PO43- group.

”Thank you for the observation, and comment, the section was modify as follows:

Fourier transform infrared (FT-IR) spectra were obtained at a wavelength range or 400-4000 cm-1 in the transmittance mode an FT-IR JASCO 4000 spectrometer, using palletized samples prepared with 5 mg of powder sample and 200 mg of KBr.  

36) Results and Discussion, line 182, p4: Results and discussion shall be separated.

Thank you for your comment, I do appreciate very much, however for this special issue on Hydrothermal publication, the Results and Discussion can be included together. There is not restrictions. Thank you very much for your comprehension.

37) Effect of Si4+ saturation on the hydrothermal synthesis of Si-Hap, lines 198-203, p5: It was unclear how reproducible was the XRD patterns in “The sample prepared with the 0.3 M TMAS solution with the volume to supply 0.33 mol% Si exhibited a slight shifting of the peak to a lower diffraction angle (Figure 1a). ). In contrast, at 4.5 and 9-fold saturation levels, a progressive displacement of the XRD pattern proceeded at small 2θ angles, and also a remarkable peak broadening occurred on the diffraction patterns, as shown in Figures 1(b-c).”

Thank you very much again for the comments: It was correct in the way you have suggested:

First of all, the samples were prepared by duplicate, also all the XRD was determined with the same XRD equipment and all the patterns were obtained at the same conditions. In such a case It was enough reproducible.

Furthermore, for the sample prepared with 0.3M of TMAS solutions slight shifting of the peak to a lower diffraction angle, its means that, the low concentration of Si4+ did not change the cell on the HAp structure. In contrast, large Si4+ content, promote a remarkable shifting to low 2Θ angle. 

38) Crystalline structural and chemical compositional analyses of Si-HAp powders, lines 220-228, p5: It was unclear what was the sample variations to support “The Rietveld refinement approach provided the lowest values of the goodness-of-fit factor (GOF/Chi2), averaging 1.23 ± 0.6 and low Rwp 7.32± 1.0 values (Table 1), which confirm the excellent accuracy of the refinement approach investigated in our case. Figures 2(a-b) show the calculated and experimental diffraction profiles together with the residual fitting curve obtained. The lower trace is the difference between the observed and calculated patterns depicted by the residual straight line, and the vertical lines correspond to the calculated Bragg peaks positions. It is worth mentioning the good agreement between both.”

Thank you for your kind observations and comments. It was correct in the way you have suggested:

The Rietveld refinement of Si-HA powders was conducted with the refinement algorithm includes the Si4+ and P5+ molar contents determined by wet chemical analyses (Table 1, ICP results), together with the atom occupation, as suggested elsewhere (Figure 2) [1,2,5,8]. Various crystalline structural features, were included as the refinement parameters, such as background, lattice parameters, scale factor, profile half width, crystallite size, local strain, thermal isotropic vectors, and spatial coordinates. The parameters selected provided high accuracy for calculating the structural features of the Si-HAp powders. The Rietveld refinement approach provided the lowest values of the goodness-of-fit factor (GOF/X2), averaging 1.23 ± 0.6 and low Rwp7.32± 1.0 values were obtained (Table 1), which confirm the fine structure of all Si-HAp with excellent accuracy. The calculate diffraction profiles are good agreement with the experimental ones due the small residual difference between the observed and calculated patterns depicted by the residual straight line, and the vertical lines correspond to the calculated Bragg peaks positions ( Figure 2) [8].

39) Crystalline structural and chemical compositional analyses of Si-HAp powders, Table 1, Lines 252-253, p6: Indicate number of independent measurements, standard errors TMA concentrations, and specify or detete MM62, MM66, MM63, MM50, MM51, MM53, MM64, MM68, MM67 in “Table 1. Chemical and physical features of Si-HAp powders prepared under hydrothermal assisted microwave treatments at 150 °C, 1 h using different concentrations of TMAS solutions and tripolyphosphate as P precursor.”

Thanks again for your comment; It was correct in the way you have suggested:

(* identification samples MM62,MM66, MM63; MM50, MM51, MM53; MM64, MM68 and MM65 correspond to the SI-HAp with 5,10 and 20 mol % with concentration 0.3, 09 and 1.8M respectively)

On Table 1 on the second column it was added) in the first row “Molar Concentration” (± 0.001)

40) Crystalline structural and chemical compositional analyses of Si-HAp powders,   Lines 259-261, p7: What was the reproducibility in the intensity of the OH group since it was unclear if this can also be associated to water, and specify vibrational mode located at 631 cm-1 to support “Other crystalline features of the Si-HAp powders were examined by FT-IR analyses. 259 Figures 3(a-c) show the results of the analysis for HAp and Si-HAp powders prepared 260 under the microwave-assisted hydrothermal technique, which reveals hydroxyl (OH-) 261 groups stretching (3571 cm-1) and vibration (631 cm-1) bands.”

Thanks again for your comments and question: It was correct in the way you have suggested: the reference is added.

Normally the OH signals associate to HAp structure appears as a vibrational symmetrical band (stretching) at 3571 cm-1 and the confirmation  a wake band of bending OH-  at 631 cm-1  and its changing is associated to the incorporating of Si4+ in the HAp lattice, as was reported on ref [26]. 

In addition, the adsorbed water its relatively wide; broad H-O-H, is in the range 3600–2600 cm-1).

41) Crystalline structural and chemical compositional analyses of Si-HAp powders, Lines 277-281, p7: Delete “Additionally, the intensity of the OH- group band on the 0.16 and 0.90 mol% Si samples was similar, and this was irrespective of the saturation contents of Si+4 provided by the TMAS solutions of 0.3 and 0.9 M. On the contrary, the OH- stretching band at 3571 cm-1 and 960 cm-1 is markedly reduced in Si-HAp powders prepared with the highest concentration 1.8 M of TMAS (Figure 3c).

Thank you very much for your comment, that are useful to improve this manuscript: the paragraph was change as following: And we think it will more understandable now.

Additionally, the intensity of the OH- group band on the 0.16 and 0.90 mol% Si samples was similar irrespective of the saturation contents of Si+4. On the contrary, the OH- stretching symmetrical band at 3571 cm-1 and bending OH631cm-1  markedly deceased in its absorbance in Si-HAp powders prepared with the highest concentration 1.8 M of TMAS (Figure 3c). XRD signals and FT-IT spectra confirm that the bulk Si should be incorporated in the apatite structure rather than the partially exist at the particles surface [26].

42) Crystalline structural and chemical compositional analyses of Si-HAp powders, Lines 281-284 and lines 285-287, p7: I am not convinced that the variations in XRD and IR spectra indicated solely insertion of Si into the apatite lattice. There is the possibility that suspension due to increased amount of Si induced less crystalline apatites, which could produce variable XRD and IR. The possibility that Si interacted on the surface of mineral complexes can’t be ruled out. Delete ”These structural variations together with the XRD analyses confirm that a bulky Si incorporation took place in the apatite structure rather than partially at the particle surface as reported elsewhere [26].” and “These structural results are consistent with the substitution mechanism proposed, where PO4-3 ions are replaced by SiO44− ions, causing a stoichiometric release of OH- ions, which maintains the total charge balance in the HAp structure (Equation 1).”

Thank you very much for your kind comment to improve the manuscript, we did the changes on the manuscript  as I did mention on the comments number 41 to clarify.

Additionally, the intensity of the OH- group band on the 0.16 and 0.90 mol% Si samples was similar irrespective of the saturation contents of Si+4. On the contrary, the OH- stretching symmetrical band at 3571 cm-1 and bending OH631cm-1  markedly deceased in its absorbance in Si-HAp powders prepared with the highest concentration 1.8 M of TMAS (Figure 3c). XRD signals and FT-IT spectra confirm that the bulk Si should be incorporated in the apatite structure rather than the partially exist at the particles surface [26].

With the XRD and Rietveld analysis it is confirmed the HAp and Si-HAp structure and also we identify the typical band of the bonds related to different ions OH-, PO43- and Ca2+ and Si4+  of the HAp. Also we did confirm the presence of the Si using XPS analysis.

43) Crystalline structural and chemical compositional analyses of Si-HAp powders, Lines 287-292, p7: Specify V in “Under hydrothermal conditions the saturation of Si4+ probably inducing a reduction in the amount of hydroxyl groups to compensate an extra negative electric charge produced by the incorporation of the silicate groups, and the formation of OH- vacancies might take place to maintain the charge balance neutrality, as is described by the following equation: PO43— + OH- → SiO44− + V(OH)-.”

Thank you very much for your kind comment, It was correct in the way you have suggested: to specify V:

Under hydrothermal conditions the saturation of Si4+ probably inducing a reduction in the amount of hydroxyl groups to compensate an extra negative electric charge produced by the incorporation of the silicate groups, and the formation of OH- vacancies (V) might take place to maintain the charge balance neutrality, as is described by the following equation: PO43— + OH- → SiO44− + V(OH)-. Indeed, this behaviour is consistent with the previous research work [13].

44) Crystalline structural and chemical compositional analyses of Si-HAp powders, Figure 3, Lines 294-300, p7: Replace CHby OH in each spectra located at around 1650 cm-1 “Figure 3. FT-IR analysis of the Si-HAp powders obtained by assisted hydrothermal process at 150 °C for 1 h using different 294 concentrations of [(C4H13NO5Si2) (TMAS)]: (a) 0.3 M; (b) 0.9 M and (c) 1.8 M.”

Thank you very much  for your suggestion. I do want to apologize for the mistake, Now the plots was corrected in the way you have suggested:

45) Crystalline structural and chemical compositional analyses of Si-HAp powders, Lines 303-304, p7: XRD does not indicate any information on OH groups, while IR is sensitive to water band, rephrase “Likewise, the OH- peak decreased with broadening of the band, as was mentioned elsewhere [20]; these results also support the FT-IR results and XRD results.”

Thank you so much for your kind corrections, It was correct in the way you have suggested: as following:

The OH- peak intensity decreased with broadening of the band [20]; these results are in line with those of FT-IR and XRD.

46) Crystalline structural and chemical compositional analyses of Si-HAp powders, Lines 311-315, p9 : Alternatively IR and Raman samples were not checked for reproducibility, which may explain distinct findings from single IR and Raman measurements, rephrase “However, Si-HAp samples synthesized with a large content of Si4+ (12.16 mol%) did not show the presence of the characteristic Si signal, and only small changes were detected in the vibration OH- signal at 3570 cm-1, which were slightly decreased and broadened under the high Si4+ saturation of TMAS (1.8 M) during the hydrothermal reaction.”

Thank you very much for your kind comments and the suggestion again:

According to your comment related to the reproducibility for the case of characterization by FT- IR it was obtained using the similar conditions only two times. Whilst Raman spectra only was obtained once.  So sorry, Now, It is not possible to obtain the Raman again because we have no access to the laboratory. At the moment In Mexico most of the institutions are closed due to the Covid-19 situation.

It was correct in the way you have suggested: as following:

However, the presence of silicon was very wake in powders prepared with 0.41 mol% Si, as shown in Figure 5d. In contrast, a symmetric peak corresponding to the core level Si 2p at BE of 103.3 eV was revealed in the Si-HAp samples obtained with the molar volume corresponding to 5.0 mol% Si using the TMAS solution of 0.9 M.

47) Crystalline structural and chemical compositional analyses of Si-HAp powders, Lines 311-315 and 343-353, p9: Itwas unclear if the findings were reproducible in “However, the presence of silicon was not determined in powders prepared with 0.41 336 mol% Si, as shown in Figure 5d. In contrast, a symmetric peak corresponding to the core 337 level Si 2p at BE of 103.3 eV was revealed in the Si-HAp samples obtained with the molar volume corresponding to 5.0 mol% Si using the TMAS solution of 0.9 M.” and in “ By way of contrast, experiments conducted employing the highest Si4+ saturated TMAS solution (1.8 M) exhibit a gradual uptake of Si, which was revealed by the XPS spectra in Figure 6 corresponding to HAp and Si-HAp powders prepared under the microwave-assisted hydrothermal processing. All the samples are constituted by the chemical elements that form the HAp structure. Figure 6a shows the typical doublet peak associated with the core level Ca 2p3/2 XPS BE at 347.1 eV and Ca 2p1/3 at 350.07 eV. The doublet peaks exhibited a slight increase as the silicon incorporation content increased in the synthesized samples obtained with 1.61 mol% of Si in the Si-HAp powders. Likewise, the symmetric signals of P 2p exhibited a slight increase in intensity with the increase of Si4+ content, achieving a binding energy of 132.92 eV. Furthermore, the binding energy signals for O 2p peaks are nearly symmetric (Figure 6c).”

Thank you very much for your kind comments and the suggestion.:

The XPS analysis were conducted to the SiHAp samples prepared in both concentrations ‘samples of TMAS, 09M and 1.8M.

However, the presence of silicon was very wake in powders prepared with 0.41 mol% Si, as shown in Figure 5d. In contrast, a symmetric peak corresponding to the core level Si 2p at BE of 103.3 eV was revealed in the Si-HAp samples obtained with the molar volume corresponding to 5.0 mol% Si using the TMAS solution of 0.9 M.

Whilst, the XPS spectra in Figure 6 indicated HAp and Si-HAp powders prepared in the presence of the highest Si4+ saturated TMAS solution (1.8 M) exhibit a gradual uptake of Si. All the samples are constituted by the chemical elements that form the HAp structure. Figure 6a shows the typical doublet peak associated with the core level Ca 2p3/2 XPS BE at 347.1 eV and Ca 2p1/3 at 350.07 eV. The doublet peaks increased slightly as the silicon incorporation content increased in the synthesized samples obtained with 1.61 mol% of Si in the Si-HAp powders. Likewise, the symmetric signals of P 2p increase slightly with the increase of Si4+ content, achieving a binding energy of 132.92 eV. Furthermore, the binding energy signals for O 2p peaks are nearly symmetric (Figure 6c).

48) Morphological aspects of the partially substituted Si-HAp particles prepared hydrothermally, lines 400-401, p11: Add statistically analysis of diameter sizes of samples to support “These results are supported by variation in the crystallite size, as calculated in the Rietveld refinement results (Table 1).”

Thank you very much for your kind comments and the suggestion

On table 1.  From the Rietveld analysis; the main ccrystallographic data conforms a subroutine of the program algorithm designed to conduct the structural refinement. The algorithm is based in a 10-coefficient shifted Chebyshev polynomial function for modelling the background, and a pseudo-Voigt function fitted the profile peak shape. The refinement approach calculates the unit lattice cell parameters, the isotropic thermal displacement, the crystallite size and each secondary phase content. Rietveld refinement algorithm calculated the content of each phase identified, and the schematic results are shown in table 1. On the column of the crystallite size in parentheses you will found the standard deviation of the calculate crystallite size. Normally the program its run 10 times and automatically give the average.

49) Conclusion, lines 465-466, p13: It was unclear how % of Si was determined in “The maximum amount of Si4+ incorporated in the HAp structure was 12.16 mol%, using an excess of 1.8 M of TMAS.”

Thank you very much for your kind comments: 

Certainly, under hydrothermal performed conditions mentioned in first sentence of this section, the maximum molar % of Si4+ into the HAp structure we found was 12.16 mol %, this value were determined by wet quantitative ICP analysis as was described on characterization section.

50) Conclusion, lines 469-474 , p13: It was unclear if the increased concentration of tetramethylammonium instead of Si moiety induced agglomerates in “The Si4+ excess in the reaction media caused the rod-like crystal self-assembly to produce irregular oval-shaped Si-HAp agglomerates, which were prepared under fast kinetic reaction conditions assisted by the microwave heating and exhibit sizes between 233.5 and 315.1 nm. These agglomerates exhibited a marked size coarsening, which was triggered by the Si4+ saturation level supplied in the hydrothermal media.”

Thank you very much for your kind comments again: We have proceeded according to this comment :

Certainly we found that the used excess of Si4+ in the reaction media led to the rod-like crystal self-assembly to produce irregular oval-shaped Si-HAp agglomerates, which were prepared under fast kinetic reaction conditions assisted by the microwave heating 150ºC, 1h) and exhibit sizes between 235.5± 29.7and 297.4±19.4 nm.

So, under the hydrothermal conditions we found that an excess of Si4+ can promote two behavior:

  1. The particle size decreased and
  2. The formation of agglomerates took place with the presence of large amount of Si, because the reduction of particle size itself.

51) Conclusion, lines 478-480, and lines 481-484, p13: Delete “Furthermore, a remarkable decrease on the OH- ions content was confirmed by FTIR and XPS analyses, the gradual OH- lost was caused to compensate the partial incorporation of SiO44- at tetrahedral PO43- sites in the HAp structure.” And “The present hydrothermal microwave assisted method has delivered high processing efficiency to crystallize Si-HAp particles with a control on the Si4+ content. This method has potential for processing Si-HAp bioceramic implants in medicine.”

Thank you very much for your kind comments again: according to this comment :

We consider that the OH- ions peaks were slightly decreasing as the Si4+ content increase in the powders for the samples synthesized, as it was event on FT-IR spectra, and was confirm by Raman when was clear  particularly for the samples prepare with 0.9M. another confirmation was evident oh XPS analysis for the O 1score level peak, that was deconvoluted into two peaks, and a small peak, which fits the shoulder between 532 and 534 eV, was detected.

The deconvolution indicates that the peak average BE energy was of 532.45 eV, and this peak is associated with the SiO4 units in the prepared powder samples, where the gradual increase in peak intensity in the samples synthesized with different Si4+ contents supports our inference (Figure 5c).

And the second large peak, deconvoluted at an average BE of 530.8 eV, corresponded to the O 1s core level of PO43- tetrahedral units. 

That is why we did not consider to omit this part of the conclusion.

Furthermore, a remarkable decrease on the OH- ions content was confirmed by FTIR and XPS analyses, the gradual OH- lost was caused to compensate the partial incorporation of SiO44- at tetrahedral PO43- sites in the HAp structure. The present hydrothermal microwave assisted method has delivered high processing efficiency to crystallize Si-HAp particles with a control on the Si4+ content. This method has potential for processing Si-HAp bioceramic implants in medicine.

Finally: according to your comment that: It was unclear if Si was inserted into the lattice of of apatites or was adsorbed on the surface of apatites.

In the present research  from the FT-IR, Raman and XPS  results

I thin spite of the presence of signals relate to Si was not evident on Raman  analysis, the results of the FT-IR-spectra shows clearly that  the Si was inserted inside Hexagon structure of the HAP because clearly  “The  presence of shoulder peak at 897 cm-1, and this band is assigned to the Si-O-Si vibration mode for tetrahedral SiO44- groups”.

Also, “A marked distortion of the shoulder peak together with the signals n1 and n3 of the P-O-P and PO43-major bands was found with progressively increasing the SiO44− molar content in the Si-HAp powders, as reported previously [24,26].

In addition, XPS is an extremely sensitive technique for detecting the chemical elements constituting the outermost layer of a surface up to ca. 200  . In our particular case we found that  a symmetric peak corresponding to the core level Si 2p at BE of 103.3 eV was revealed in the Si-HAp samples obtained with the molar volume corresponding to 5.0 mol% Si using the TMAS solution of 0.9 M. So,  it means that the Si was inserted into the HAp , particularly  for the samples prepared with TMAS with concentrations > 0.9M and 5.0 mol %.

Thank very much for your time and all the suggestion that were consider valuable to improve the manuscript

Round 2

Reviewer 1 Report

I found that the manuscript has been improved in comparison to the previous one. The authors have adequately replied to my comments. 

Before acceptance, please consider the following suggestions:

Figures - Please inform and use "a.u." for arbitrary units.

Please improve the quality of Figure 3; some descriptors in the plots are very hard to read as they are very small.

Figure 8. Please use the same format for scale bars.

References should be cited in text by number in order of appearance. 

Reviewer 2 Report

After reading the responses to my comments in the review and the revised version of the manuscript, I believe that it can be published in Nanomaterials

Reviewer 3 Report

General comments:

The authors addressed several of my concerns but there are still significant concerns that were insufficiently addressed, see points 1), 3), 17), 23), 25), 37), 38), 39), 40), 41), 46), 47), 48), 50), and 51).

1) Replace hydroxyapatites by apatites in “The synthesis of partially substituted silicon hydroxyapatite (Si-HAp) nanopowders was systematically investigated via the microwave-assisted hydrothermal process.”

Thank you very much for the time spent to review our manuscript. We found all the comments useful to improve the fundamentals in the document, and we entirely address all the comments made to our preliminary manuscript version.

In the present research it’s been preparing synthetic Substitute Silicon hydroxyapatite and into the hexagonal structure still content OH- ion inside the structure Ca10(PO43-)6-x(SiO44-)x(OH)2-x(s).

In many papers: Hydroxyapatite, it is called hydroxyapatite (HAp or HA), for naturally occurring mineral form of calcium apatite with the formula Ca5(PO4)3(OH), but is usually written Ca10(PO4)6(OH)2 to denote the crystal unit cell .

Normally, it’s found that the term of Apatite is more common for natural and biological apatite rather than synthetic hydroxyapatite, that is why we did used the term of Ca10(PO4)6(OH)2 “Hydroxyapatite”.

Answer to the authors: As pointed by the authors the chemical formula of hydroxyapatite (HA)  is Ca10(PO4)6(OH)2. Therefore Ca10(PO43-)6-x(SiO44-)x(OH)2-x(s) can’t be called HA, but apatites.

3) Introduction, lines 35-37, p1: Specify HAp in the main text, replace hydroxyapatites by apatites, and rephrase “The preparation of biomaterials with similar characteristics to biological hydroxyapatite, in terms of their chemical and physical properties, involves the uptake of 36 cations and anions in the hexagonal HAp structure.”

Thank you very much for the suggestion, It was modify in manuscript as follows:

The preparation of biomaterials with similar chemical and physical properties to biological hydroxyapatite (HAp), in terms of their chemical and physical properties, involves the uptake of cations and anions in the hexagonal HAp structure. The incorporation of Si4+ ions into the PO43- unit network of the HAp stimulates both bone formation and resorption processes, which are relevant to both tissue restauration and bone growth [1].

Answer to the authors: There are no hydroxyapatites in vivo. Biological Hap is misleading since  biological apatite contain other types of anions than OH-.  Replace biological hydroxyapatite (HAP) by apatite.

17) Introduction, lines 80-85, p2: Apatites are difficult to be formed at high pH, delete or rephrase, and add references to support “The low silicon reactivity in the hydrothermal alkaline medium at a pH of 10 caused a limited Si4+ content in the HAp structure of 30 mol% regarding the stoichiometric amount selected (1–20 mol% Si). In this case, the incorporation of CO32- ions was not the cause of the significant Si uptake. The high solubility of the TMAS in the alkaline solution is likely to produce Si complex ions that are highly stable in the hydrothermal medium.”

Thank you for your comment and It was correct in the way you have suggested:

The low silicon reactivity in the hydrothermal alkaline medium at a pH of 10 caused a limited Si4+ content in the HAp structure of 30 mol% regarding the stoichiometric amount selected (1–20 mol% Si). In this case, the incorporation of CO32- ions was not the cause of the significant Si uptake. The high solubility of the TMAS in the alkaline solution is likely to produce Si complex ions that are highly stable in the hydrothermal medium, giving rise to the decrease in the Si concentration in the embryo and growth steps [20].

Answer to the authors: It is still unclear why apatites were prepared under alkaline conditions and not under neutral conditions. Some explanations and references are needed.

23) Materials, lines 112-114, p3: How the stoichiometry was controlled under saturated condition? in “This reagent was used to produce three different solutions with concentrations of 0.3, 0.9 and 1.8 M, which are saturated in Si4+ in comparison with the stoichiometric of 0.2 M.”

Thank you for your comment and It was correct in the way you have suggested

In line 110- 111; The stoichiometric is controlling because initial Mix of the all starting reagents was adjusted the pH to 10 and immediately the chamber of the autoclaves were hermetically closed and after that was conducted the hydrothermal treatment .

Answer to the authors: It is still unclear if the there were aggregates to impact the 0.2M concentration since there were saturated solutions.

25) Microwave-assisted hydrothermal synthesis, lines 118-119, p3: How the stoichiometry was controlled under saturated condition?, specify % in what medium In ” The silicon content selected to produce Ca10(PO4)6-x(SiO4)x(OH)2-x was 6, 10 and 20 118 mol%.

Thank you very much for your kind question: The answer is as follows:

Because the reaction system is in aqueous solution; initially it was fix the initial stoichiometric of the reaction by calculate the appropriated amount of each starting reagent and adjusted the initial pH in 10. For all the experiments the pH was checked at the initial and the final steps to ensure the appropriated control of the alkali in the aqueous medium during the reaction.

Answer to the authors: The authors shall add something like “The stoichiometry corresponded to the initial concentrations of components and do not necessarily reflect the final stoichiometry since there were no elemental analysis.”

37) Effect of Si4+ saturation on the hydrothermal synthesis of Si-Hap, lines 198-203, p5: It was unclear how reproducible was the XRD patterns in “The sample prepared with the 0.3 M TMAS solution with the volume to supply 0.33 mol% Si exhibited a slight shifting of the peak to a lower diffraction angle (Figure 1a). ). In contrast, at 4.5 and 9-fold saturation levels, a progressive displacement of the XRD pattern proceeded at small 2θ angles, and also a remarkable peak broadening occurred on the diffraction patterns, as shown in Figures 1(b-c).”

Thank you very much again for the comments: It was correct in the way you have suggested:

First of all, the samples were prepared by duplicate, also all the XRD was determined with the same XRD equipment and all the patterns were obtained at the same conditions. In such a case It was enough reproducible.

Furthermore, for the sample prepared with 0.3M of TMAS solutions slight shifting of the peak to a lower diffraction angle, its means that, the low concentration of Si4+ did not change the cell on the HAp structure. In contrast, large Si4+ content, promote a remarkable shifting to low 2Θ angle. 

Answer to the authors: Duplicates are insufficient to insure reproducibility of experiments. At least three independent experiments shall be performed (and not duplicates) to correctly assess the reproducibility of the preparation. It was unclear if there were variations during the preparations.

38) Crystalline structural and chemical compositional analyses of Si-HAp powders, lines 220-228, p5: It was unclear what was the sample variations to support “The Rietveld refinement approach provided the lowest values of the goodness-of-fit factor (GOF/Chi2), averaging 1.23 ± 0.6 and low Rwp 7.32± 1.0 values (Table 1), which confirm the excellent accuracy of the refinement approach investigated in our case. Figures 2(a-b) show the calculated and experimental diffraction profiles together with the residual fitting curve obtained. The lower trace is the difference between the observed and calculated patterns depicted by the residual straight line, and the vertical lines correspond to the calculated Bragg peaks positions. It is worth mentioning the good agreement between both.”

Thank you for your kind observations and comments. It was correct in the way you have suggested:

The Rietveld refinement of Si-HA powders was conducted with the refinement algorithm includes the Si4+and P5+ molar contents determined by wet chemical analyses (Table 1, ICP results), together with the atom occupation, as suggested elsewhere (Figure 2) [1,2,5,8]. Various crystalline structural features, were included as the refinement parameters, such as background, lattice parameters, scale factor, profile half width, crystallite size, local strain, thermal isotropic vectors, and spatial coordinates. The parameters selected provided high accuracy for calculating the structural features of the Si-HAp powders. The Rietveld refinement approach provided the lowest values of the goodness-of-fit factor (GOF/X2), averaging 1.23 ± 0.6 and low Rwp7.32± 1.0 values were obtained (Table 1), which confirm the fine structure of all Si-HAp with excellent accuracy. The calculate diffraction profiles are good agreement with the experimental ones due the small residual difference between the observed and calculated patterns depicted by the residual straight line, and the vertical lines correspond to the calculated Bragg peaks positions ( Figure 2) [8].

Answer to the authors:  My concern was on the reproducibility of the values. What was the sample variation during distinct experiments? And not on one single measurement.

39) Crystalline structural and chemical compositional analyses of Si-HAp powders, Table 1, Lines 252-253, p6: Indicate number of independent measurements, standard errors TMA concentrations, and specify or detete MM62, MM66, MM63, MM50, MM51, MM53, MM64, MM68, MM67 in “Table 1. Chemical and physical features of Si-HAp powders prepared under hydrothermal assisted microwave treatments at 150 °C, 1 h using different concentrations of TMAS solutions and tripolyphosphate as P precursor.”

Thanks again for your comment; It was correct in the way you have suggested:

(* identification samples MM62,MM66, MM63; MM50, MM51, MM53; MM64, MM68 and MM65 correspond to the SI-HAp with 5,10 and 20 mol % with concentration 0.3, 09 and 1.8M respectively)

On Table 1 on the second column it was added) in the first row “Molar Concentration” (± 0.001)

Answer to the authors:  The number of independent measurements was not mentioned.

40) Crystalline structural and chemical compositional analyses of Si-HAp powders, Lines 259-261, p7: What was the reproducibility in the intensity of the OH group since it was unclear if this can also be associated to water, and specify vibrational mode located at 631 cm-1 to support “Other crystalline features of the Si-HAp powders were examined by FT-IR analyses. 259 Figures 3(a-c) show the results of the analysis for HAp and Si-HAp powders prepared 260 under the microwave-assisted hydrothermal technique, which reveals hydroxyl (OH-) 261 groups stretching (3571 cm-1) and vibration (631 cm-1) bands.”

Thanks again for your comments and question: It was correct in the way you have suggested: the reference is added.

Normally the OH signals associate to HAp structure appears as a vibrational symmetrical band (stretching) at 3571 cm-1 and the confirmation a wake band of bending OH- at 631 cm-1 and its changing is associated to the incorporating of Si4+ in the HAp lattice, as was reported on ref [26]. 

In addition, the adsorbed water its relatively wide; broad H-O-H, is in the range 3600–2600 cm-1).

Answers to the authors: It was insufficiently answered. It is still unclear how IR spectra varied in the OH range, especially that it seems that there were no attempts to determine IR spectra from distinct batches of preparations to check reproducibility of IR spectra.  

41) Crystalline structural and chemical compositional analyses of Si-HAp powders, Lines 277-281, p7: Delete “Additionally, the intensity of the OH- group band on the 0.16 and 0.90 mol% Si samples was similar, and this was irrespective of the saturation contents of Si+4 provided by the TMAS solutions of 0.3 and 0.9 M. On the contrary, the OH- stretching band at 3571 cm-1 and 960 cm-1 is markedly reduced in Si-HAp powders prepared with the highest concentration 1.8 M of TMAS (Figure 3c).

Thank you very much for your comment, that are useful to improve this manuscript: the paragraph was change as following: And we think it will more understandable now.

Additionally, the intensity of the OH- group band on the 0.16 and 0.90 mol% Si samples was similar irrespective of the saturation contents of Si+4. On the contrary, the OH- stretching symmetrical band at 3571 cm-1 and bending OH-  631cm-1 markedly deceased in its absorbance in Si-HAp powders prepared with the highest concentration 1.8 M of TMAS (Figure 3c). XRD signals and FT-IT spectra confirm that the bulk Si should be incorporated in the apatite structure rather than the partially exist at the particles surface [26].

Answers to the authors: No conclusions can be reached at this stage due to lack of reproducibility of IR spectra.

46) Crystalline structural and chemical compositional analyses of Si-HAp powders, Lines 311-315, p9 : Alternatively IR and Raman samples were not checked for reproducibility, which may explain distinct findings from single IR and Raman measurements, rephrase “However, Si-HAp samples synthesized with a large content of Si4+ (12.16 mol%) did not show the presence of the characteristic Si signal, and only small changes were detected in the vibration OH- signal at 3570 cm-1, which were slightly decreased and broadened under the high Si4+ saturation of TMAS (1.8 M) during the hydrothermal reaction.”

Thank you very much for your kind comments and the suggestion again:

According to your comment related to the reproducibility for the case of characterization by FT- IR it was obtained using the similar conditions only two times. Whilst Raman spectra only was obtained once. So sorry, Now, It is not possible to obtain the Raman again because we have no access to the laboratory. At the moment In Mexico most of the institutions are closed due to the Covid-19 situation.

It was correct in the way you have suggested: as following:

However, the presence of silicon was very wake in powders prepared with 0.41 mol% Si, as shown in Figure 5d. In contrast, a symmetric peak corresponding to the core level Si 2p at BE of 103.3 eV was revealed in the Si-HAp samples obtained with the molar volume corresponding to 5.0 mol% Si using the TMAS solution of 0.9 M.

Answers to the authors:  One single Raman spectrum is insufficient to assess firmly the findings. The Raman spectrum and its relative comments shall be deleted.

47) Crystalline structural and chemical compositional analyses of Si-HAp powders, Lines 311-315 and 343-353, p9: Itwas unclear if the findings were reproducible in “However, the presence of silicon was not determined in powders prepared with 0.41 336 mol% Si, as shown in Figure 5d. In contrast, a symmetric peak corresponding to the core 337 level Si 2p at BE of 103.3 eV was revealed in the Si-HAp samples obtained with the molar volume corresponding to 5.0 mol% Si using the TMAS solution of 0.9 M.” and in “ By way of contrast, experiments conducted employing the highest Si4+ saturated TMAS solution (1.8 M) exhibit a gradual uptake of Si, which was revealed by the XPS spectra in Figure 6 corresponding to HAp and Si-HAp powders prepared under the microwave-assisted hydrothermal processing. All the samples are constituted by the chemical elements that form the HAp structure. Figure 6a shows the typical doublet peak associated with the core level Ca 2p3/2 XPS BE at 347.1 eV and Ca 2p1/3 at 350.07 eV. The doublet peaks exhibited a slight increase as the silicon incorporation content increased in the synthesized samples obtained with 1.61 mol% of Si in the Si-HAp powders. Likewise, the symmetric signals of P 2p exhibited a slight increase in intensity with the increase of Si4+ content, achieving a binding energy of 132.92 eV. Furthermore, the binding energy signals for O 2p peaks are nearly symmetric (Figure 6c).”

Thank you very much for your kind comments and the suggestion.:

The XPS analysis were conducted to the SiHAp samples prepared in both concentrations ‘samples of TMAS, 09M and 1.8M.

However, the presence of silicon was very wake in powders prepared with 0.41 mol% Si, as shown in Figure 5d. In contrast, a symmetric peak corresponding to the core level Si 2p at BE of 103.3 eV was revealed in the Si-HAp samples obtained with the molar volume corresponding to 5.0 mol% Si using the TMAS solution of 0.9 M.

Whilst, the XPS spectra in Figure 6 indicated HAp and Si-HAp powders prepared in the presence of the highest Si4+ saturated TMAS solution (1.8 M) exhibit a gradual uptake of Si. All the samples are constituted by the chemical elements that form the HAp structure. Figure 6a shows the typical doublet peak associated with the core level Ca 2p3/2 XPS BE at 347.1 eV and Ca 2p1/3 at 350.07 eV. The doublet peaks increased slightly as the silicon incorporation content increased in the synthesized samples obtained with 1.61 mol% of Si in the Si-HAp powders. Likewise, the symmetric signals of P 2p increase slightly with the increase of Si4+ content, achieving a binding energy of 132.92 eV. Furthermore, the binding energy signals for O 2ppeaks are nearly symmetric (Figure 6c).

Answers to the authors: The reproducibility of the experiments was not addressed.

48) Morphological aspects of the partially substituted Si-HAp particles prepared hydrothermally, lines 400-401, p11: Add statistically analysis of diameter sizes of samples to support “These results are supported by variation in the crystallite size, as calculated in the Rietveld refinement results (Table 1).”

Thank you very much for your kind comments and the suggestion

On table 1. From the Rietveld analysis; the main ccrystallographic data conforms a subroutine of the program algorithm designed to conduct the structural refinement. The algorithm is based in a 10-coefficient shifted Chebyshev polynomial function for modelling the background, and a pseudo-Voigt function fitted the profile peak shape. The refinement approach calculates the unit lattice cell parameters, the isotropic thermal displacement, the crystallite size and each secondary phase content. Rietveld refinement algorithm calculated the content of each phase identified, and the schematic results are shown in table 1. On the column of the crystallite size in parentheses you will found the standard deviation of the calculate crystallite size. Normally the program its run 10 times and automatically give the average.

Answers to the authors: The statistical analysis was insufficient since it seems to be based from one single experiment. At least three independent experiments shall be performed to obtain reliable statistics.

50) Conclusion, lines 469-474 , p13: It was unclear if the increased concentration of tetramethylammonium instead of Si moiety induced agglomerates in “The Si4+ excess in the reaction media caused the rod-like crystal self-assembly to produce irregular oval-shaped Si-HAp agglomerates, which were prepared under fast kinetic reaction conditions assisted by the microwave heating and exhibit sizes between 233.5 and 315.1 nm. These agglomerates exhibited a marked size coarsening, which was triggered by the Si4+ saturation level supplied in the hydrothermal media.”

Thank you very much for your kind comments again: We have proceeded according to this comment :

Certainly we found that the used excess of Si4+ in the reaction media led to the rod-like crystal self-assembly to produce irregular oval-shaped Si-HAp agglomerates, which were prepared under fast kinetic reaction conditions assisted by the microwave heating 150ºC, 1h) and exhibit sizes between 235.5± 29.7and 297.4±19.4 nm.

So, under the hydrothermal conditions we found that an excess of Si4+ can promote two behavior:

  1. The particle size decreased and
  2. The formation of agglomerates took place with the presence of large amount of Si, because the reduction of particle size itself.

Answers to the author: It was insufficiently addressed. It is still unclear whether if tetramethlyl  ammonium induced aggregations

51) Conclusion, lines 478-480, and lines 481-484, p13: Delete “Furthermore, a remarkable decrease on the OH- ions content was confirmed by FTIR and XPS analyses, the gradual OH- lost was caused to compensate the partial incorporation of SiO44- at tetrahedral PO43- sites in the HAp structure.” And “The present hydrothermal microwave assisted method has delivered high processing efficiency to crystallize Si-HAp particles with a control on the Si4+ content. This method has potential for processing Si-HAp bioceramic implants in medicine.”

Thank you very much for your kind comments again: according to this comment :

We consider that the OH- ions peaks were slightly decreasing as the Si4+ content increase in the powders for the samples synthesized, as it was event on FT-IR spectra, and was confirm by Raman when was clear particularly for the samples prepare with 0.9M. another confirmation was evident oh XPS analysis for the O 1score level peak, that was deconvoluted into two peaks, and a small peak, which fits the shoulder between 532 and 534 eV, was detected.

The deconvolution indicates that the peak average BE energy was of 532.45 eV, and this peak is associated with the SiO4 units in the prepared powder samples, where the gradual increase in peak intensity in the samples synthesized with different Si4+ contents supports our inference (Figure 5c).

And the second large peak, deconvoluted at an average BE of 530.8 eV, corresponded to the O 1s core level of PO43- tetrahedral units. 

That is why we did not consider to omit this part of the conclusion.

Furthermore, a remarkable decrease on the OH- ions content was confirmed by FTIR and XPS analyses, the gradual OH- lost was caused to compensate the partial incorporation of SiO44- at tetrahedral PO43- sites in the HAp structure. The present hydrothermal microwave assisted method has delivered high processing efficiency to crystallize Si-HAp particles with a control on the Si4+ content. This method has potential for processing Si-HAp bioceramic implants in medicine.

Finally: according to your comment that: It was unclear if Si was inserted into the lattice of of apatites or was adsorbed on the surface of apatites.

In the present research from the FT-IR, Raman and XPS results

I thin spite of the presence of signals relate to Si was not evident on Raman analysis, the results of the FT-IR-spectra shows clearly that the Si was inserted inside Hexagon structure of the HAP because clearly “The presence of shoulder peak at 897 cm-1, and this band is assigned to the Si-O-Si vibration mode for tetrahedral SiO44- groups”.

Also, “A marked distortion of the shoulder peak together with the signals n1 and n3 of the P-O-P and PO43-major bands was found with progressively increasing the SiO44− molar content in the Si-HAp powders, as reported previously [24,26].

In addition, XPS is an extremely sensitive technique for detecting the chemical elements constituting the outermost layer of a surface up to ca. 200 . In our particular case we found that a symmetric peak corresponding to the core level Si 2p at BE of 103.3 eV was revealed in the Si-HAp samples obtained with the molar volume corresponding to 5.0 mol% Si using the TMAS solution of 0.9 M. So, it means that the Si was inserted into the HAp , particularly for the samples prepared with TMAS with concentrations > 0.9M and 5.0 mol %.

Answers to the authors: It was insufficiently addressed. The problem lies in the reproducibility from different sets of experiments. At this stage, it seems that single experiment with eventually duplicates were performed. Duplicates are insufficient to firmly assess the reproducibility of the findings, at least three independent experiments shall be performed.